# 3D molecular phenotyping of cleared human brain tissues with light-sheet fluorescence microscopy

Luca Pesce [1,2], Marina Scardigli[1,2], Vladislav Gavryusev [1,2], Annunziatina Laurino[1,7], Giacomo Mazzamuto [1,3], Niamh Brady[1], Giuseppe Sancataldo[1], Ludovico Silvestri [1,2,3], Christophe Destrieux[4], Patrick R. Hof[5], Irene Costantini [1,3,6 ✉] & Francesco S. Pavone[1,2,3]

The combination of optical tissue transparency with immunofluorescence allows the molecular characterization of biological tissues in 3D. However, adult human organs are particularly challenging to become transparent because of the autofluorescence contributions of aged tissues. To meet this challenge, we optimized SHORT (SWITCH—$H_2O_2$—antigen Retrieval—TDE), a procedure based on standard histological treatments in combination with a refined clearing procedure to clear and label portions of the human brain. 3D histological characterization with multiple molecules is performed on cleared samples with a combination of multi-colors and multi-rounds labeling. By performing fast 3D imaging of the samples with a custom-made inverted light-sheet fluorescence microscope (LSFM), we reveal fine details of intact human brain slabs at subcellular resolution. Overall, we proposed a scalable and versatile technology that in combination with LSFM allows mapping the cellular and molecular architecture of the human brain, paving the way to reconstruct the entire organ.

[1] European Laboratory for Non-linear Spectroscopy (LENS), University of Florence, Sesto Fiorentino, Italy. [2] Department of Physics and Astronomy, University of Florence, Sesto Fiorentino, Italy. [3] National Institute of Optics (INO), National Research Council (CNR), Sesto Fiorentino, Italy. [4] UMR 1253, iBrain, Université de Tours, Tours, France. [5] Nash Family Department of Neuroscience and Friedman Brain Institute, Icahn School of Medicine at Mount Sinai, New York, NY, USA. [6] Department of Biology, University of Florence, Florence, Italy. [7] Present address: Department of Neurosciences, Psychology, Drug Research and Child Health (NEUROFARBA), University of Florence, Florence, Italy. ✉email: costantini@lens.unifi.it

Volumetric reconstruction with a cellular resolution of samples, such as the human brain, is difficult to obtain due to the geometrical, structural, and biological composition of the organ itself. To overcome some of these issues, sections with a thickness less than 100 µm are used to investigate the human cortical cytoarchitecture[1,2]. However, this strategy lacks three-dimensionality and can decontextualize the section from the surrounding environment, introducing artifacts. Several volumetric imaging technologies like confocal, multiphoton, and light sheet-fluorescence microscopy (LSFM) have been employed for visualizing deeper inside the tissue[2–6]. To perform 3D reconstructions, these optical techniques require an optimized clearing protocol, which preserves the proteins of interest and removes the components that affect the imaging (generally lipids and chromophores) by producing a refractive index mismatching that is correlated with an increase in light scattering and absorption by chromophores and pigments[3,7,8]. By removing such undesirable biological components, two main important features can be obtained: first, the refractive index can be easily matched, allowing efficient light penetration and imaging at depth; second, the absence of biological barriers like lipid biolayers allows performing deep exogenous labeling of the tissue. Although the advances in tissue clearing and labeling techniques coupled with fast optical techniques have enabled access to the structure of large intact biological tissues, translation of such methodologies to the large scale of human cerebral cortex samples has remained a challenge in neuroscience[2].

In the last few years, researchers have developed several methods for improving the clearing process and the probe penetration in large tissue specimens. Treatments like solvent dehydration[9], elasticizing tissue[10], delipidation using lipid micelles[11], denaturation[12,13], and partial digestion[14], have allowed increasing the dye diffusion in fixed samples. For these reasons, biological models like embryonic samples[15], characterized by low extracellular matrix, or thin tissue sections (<100 µm) are preferred for performing 3D staining[1,16]. Working with primate brains presents significant differences and difficulties in comparison to rodents. For example, size, high autofluorescence signal of vessels and lipopigments, high post-mortem variability, and the neuronal and myelin densities, are factors that can compromise the staining and the clearing efficiency in the human and non-human primate cerebral cortex[2]. Recent applications of optical clearing methods to the human brain provide new details for structure-function relationships in healthy and pathological conditions. However, such methodologies have some drawbacks and limitations. Most of them are compatible with specific samples such as fresh-frozen samples[17], pediatric tissue[5], fetal brains[18], and specimens with controlled post-mortem conditions[5,11,19,20]. Also, the immunostaining is usually performed with small dyes[20]/few antibodies and thinner sections[21–25]. The advent of the tissue-hydrogel engineering methodologies has extended the utility and labeling efficiency in human samples. In particular ELAST[10] converts the tissue into an elastic hydrogel allowing homogenous staining of a 1 cm-thick section with several antibodies. However, the sample preparation requires long processing times and advanced custom-made equipment. Additionally, the recent advent of expansion microscopy[26] and its variants[27] grants super-resolution imaging by expanding the hybrid sample-hydrogel[28], increasing the sample mesh size and improving the accessibility and density of labeling for proteins even within a dense specimen region[27,29,30]. However, these methodologies require strong denaturation processes, which can induce epitope damaging and alteration in the tissue architecture, making challenging the detection of multiple markers[1]. Small dye based on oligonucleotides (aptamers)[31] or nanobodies[32] can be the best choice for achieving a 3D

homogenous staining. However, the production of these probes for different epitopes[33] could be demanding and time-consuming, and the biochemical complexity of such samples may interfere with the staining process, generating a superficial probe accumulation[14]. Moreover, the inter- and intramolecular binding effects generated by formaldehyde and amino groups of proteins during a prolonged paraffin fixation could induce deep epitope changes with a masking effect[12,16,34]. Also, the time extension for achieving sufficient transparency can affect the staining efficiency, and several antibodies can fail to work following a tissue transformation protocol in the grey as well as white matter[2]. For instance, CLARITY procedures strongly reduce the staining efficiency for specific epitopes in human formalin-fixed specimens[35]. Due to these problems, large-scale reconstruction in the white and grey matter of long-storage formalin-fixed samples, with different neuronal markers and cell types with molecular details remains an unmet goal in the human brain.[12,16,34]

Based on the physicochemical properties of human brain samples, we developed a protocol compatible with specimens that show different tissue characteristics and postmortem treatments. Such optimization consists of a simple sequence of treatments that enable to clear and decolorize the sample, as well as to increase the sensitivity of reactions between antibodies and antigens in a large portion of the human brain cortex to allow high-resolution volumetric acquisition with LSFM In particular, optimal transparency and access of external probes is enhanced by the clearing process, the autofluorescence of the fixated tissue is bleached, and the unmasked effect of antigen retrieval is performed. This method, termed SHORT (SWITCH—$H_2O_2$— antigen Retrieval—TDE) employs the tissue ultrastructure preservation of the SWITCH protocol[23] with a series of conventional buffers used in the histological treatments for performing cm-size mesoscopic analysis of 500 µm-thick slices with light-sheet fluorescence microscopy (LSFM). Using our custom-made LSFM, we evaluated the staining efficiency in the white and grey matter, and we assessed the 500-µm thickness as a limit for obtaining a homogenous labeling for different antibody staining (both neuronal, glial, and vasculature). This result increases by 5 times the slice thickness treated with the classic SWITCH technique[23] allowing to reduce the cutting artifact and the acquisition times. Additionally, we optimized multiple staining and multiple round procedures to characterize with several markers the same tissue, obtaining comprehensive molecular phenotyping of the sample. Finally, we demonstrated the widespread applicability of the method by coupling SHORT with LSFM to investigate the 3D cytoarchitecture of different human brain areas (i.e., hippocampus, Broca's area, motor cortex, precentral cortex, and superior frontal cortex) at subcellular resolution and with a fast acquisition rate.

## Results

**Autofluorescence reduction characterization of cleared tissues.** To evaluate the effect of tissue clearing on human brain tissue we examined the autofluorescence (AF) signal of transformed tissues. To this aim, we selected two different tissue transformation clearing approaches: SWITCH[23] and SHIELD[25] to process human 500-µm thick slices. The family of these transformation tissue methods consists in the control of the chemical reaction via specific buffers, termed "system-wide control of interaction time and kinetics of chemicals (SWITCH)-Off buffer". In such a solution, the intramolecular and intermolecular cross-link reactions are completely suppressed, to inhibit the fixative molecules depletion (glutaraldehyde for SWITCH or polyglycerol-3-polyglycidyl for SHIELD) on the surface and diffuse them throughout the tissue. By moving the specimens to a SWITCH-On buffer, the fixative initializes a uniform and global

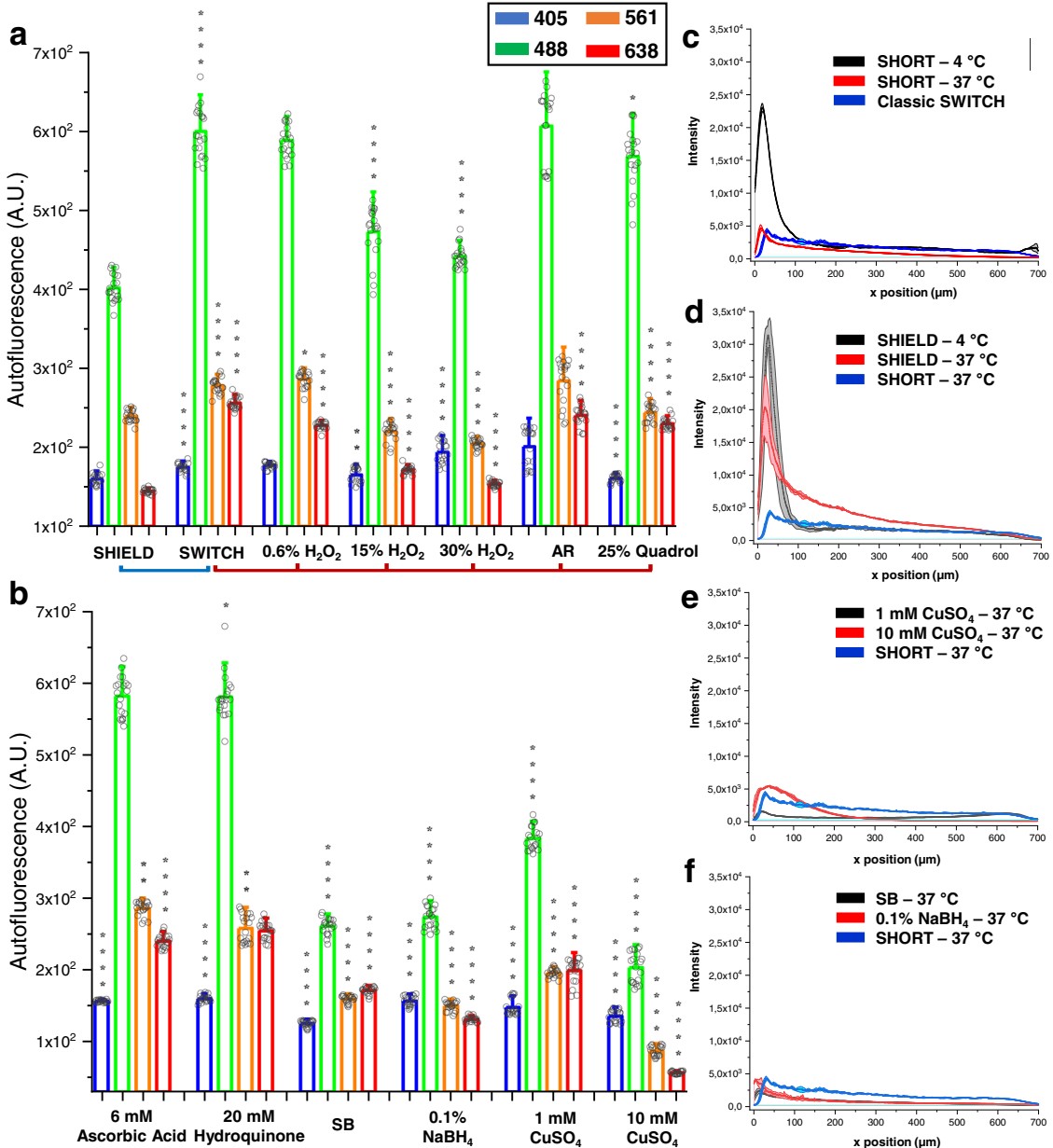

**Fig. 1 Effect of autofluorescence elimination reagents—AR and NeuN antibody penetration in processed human slices. a**, **b** Comparison of seven different autofluorescence elimination reagents and AR in 500 μm-SWITCH processed slices using LSFM with the excitation light at 405, 488, 561, and 638 nm. For $H_2O_2$ and $CuSO_4$, we examined three and two different concentrations, respectively. Statistical analysis ($n = 20$) was performed between two transformation protocol methods (SWITCH vs SHIELD) (**a**) and classic SWITCH-processed tissue vs autofluorescence elimination treatments on SWITCH-treated slices (**a**, **b**). A Mann–Whitney test was performed (*$P < 0.05$, **$P < 0.01$, ***$P < 0.001$, ****$P < 0.0001$). Error bars show mean ± SD. Abbreviation: AR antigen retrieval, SB Sudan black. **c**, **d** Effect of temperature (4 °C, 37 °C) and tissue processing (SWITCH, SHIELD, and SHORT), **e**, **f** autofluorescence treatments (SB, $NaBH_4$, and $CuSO_4$) on NeuN immunofluorescence labeling. The plot profiles in **c**–**f** show the mean intensity ± SD of three different regions.

fixation/gelation. This process transforms the tissue into a heat- and chemical-resistant hybrid, which allows rapidly clearing at high temperatures and staining with several probes, including antibodies and fluorescent dyes. The advantage of such procedures is their compatibility with post-mortem fixation of the specimens without the need for a perfusion step. The delipidated specimens were then equilibrated with TDE to reach the final transparency, as previously used[6], and examined by LSFM using four excitation wavelengths at 405, 488, 561, and 638 nm.

The natural AF at 405 and 488 nm in both protocols is likely attributable to the normal presence of the reduced form of

nicotinamide adenine dinucleotide (NAD(P)H), the oxidized form of flavins, and vessels in the human cortex[36]. However, as shown in Fig. 1a, the crosslinking agent glutaraldehyde in SWITCH induces a higher AF signal compared to SHIELD at 488 nm. Moreover, in the red-shifted spectrum, lipofuscin pigments show highly intensity signals, approximately at 561 and 638 nm, localized primarily to the neuronal soma[37] (Fig. 1a). In this spectral range, further AF signal is especially marked in the SWITCH processed tissue, and it can be imputable to the crosslink reaction between lysine residues and glutaraldehyde, which produces a strong yellow and red emission[38].

To decrease the AF introduced by the tissue transformation treatments, seven approaches, previously reported to reduce the AF of several tissue types and cells were tested: hydrogen peroxide $(H_2O_2)$[15,39], Quadrol[40–42], ascorbic acid[38], hydroquinone[38], sodium borohydride $(NaBH_4)$[43,44], copper sulfate $(CuSO_4)$[45], and Sudan Black (SD)[43,45,46]. These treatments were tested for their easy preparation combined with rapid decolorization procedures. As SHIELD showed low background autofluorescence across the visible spectral range[25], we used this clearing method as the reference standard for probing the AF decrease in SWITCH processed slices (Fig. 1a, b).

In addition to classic AF-quenching methods, we also investigated the role of antigen retrieval (AR) treatments on transformed tissues. Indeed, although AR is not implicated in the quenching process, we wanted to integrate it in the clearing procedure for its epitope-unmasking effects to increase immuno-fluorescence compatibility. In agreement with previous works[47–49], the alkaline AR solutions (including Tris-EDTA buffer) showed the highest retrieved antigen immunoreactivity against most of the antibodies tested relatively to the AR solutions at pH 6.0 (i.e. sodium citrate buffer).

We found that high-temperature AR for a short time using tris-EDTA buffer (see recipe in Methods), the reducing agent's ascorbic acid and hydroquinone showed no significant changes in the overall AF spectrum in SWITCH processed human cortex slices (Fig. 1a, b). Also, Quadrol showed a red-shifted quenching effect, while AR presented a reduction of the intensity at 638 nm, likely attributable to the reduction of lipopigments in the human cortex. Although low concentration for a short time of oxygen peroxide did not significantly modify the AF signal, 30% $H_2O_2$ significantly diminished the tissue AF at 405, 488, 561, and 638 nm excitation light (Fig. 1a). Also, $NaBH_4$, $CuSO_4$, and SB (see Autofluorescence Elimination Reagent in Methods) were the most effective autofluorescence quenchers, reducing the AF signal about twofold compared to SWITCH and SHIELD processed slices (Fig. 1b and Fig. S1). For these reasons, 30% $H_2O_2$, $NaBH_4$, and $CuSO_4$ were selected for their capability for diminishing the AF on human cortex tissue and used for the analysis of immunostaining penetration depth discussed below.

**Optimization of SHORT immunostaining**. As a prerequisite step to performing the molecular phenotyping of cleared human brain specimens, we analyzed the diffusion capability of different antibodies to select the most effective ones. After the staining, the penetration of the antibodies deep inside the sample was evaluated by imaging it with LSFM throughout the 500-μm thickness of the slice. We found that the first 200 μm shows a high fluorescence signal compared to the middle and the opposite side. Indeed, several physical processes, such as light scattering and absorption[50,51], optical aberration and the tissue properties (like low transparency) can induce signal decay and contrast loss[3].

To address this issue, we tested immunofluorescence staining on 500-μm thick SWITCH- and SHIELD-transformed tissue using different AF elimination treatments, AR, antibodies incubation temperature, and buffers. In particular, we tested several combinations of different AF elimination reagents, unmasking effects of AR (see above), temperature (37 °C and 4 °C) (Fig. 1c, d), buffers (PBST, PBST + BSA, HEPES buffer supplemented with Quadrol and urea; Fig. S2d) and antibody concentrations, to determine the conditions that permitted the best antibody penetration efficiency (Fig. S3a–f). For the analysis we used an anti-NeuN antibody (a neuron-specific nuclear protein)[52,53] to validate the protocol with a high-density epitopes staining. Results showed that the fluorescence intensity displays a rimmed pattern, marked at 4 °C, with an intense peak at 50 μm

depth in SWITCH and SHIELD (Fig. 1c, d). Instead, high-temperature incubation (37 °C) generates an almost homogenous probe distribution in human SWICH-processed slices treated with $H_2O_2$ and AR. These variations upon the original SWITCH protocol clearly demonstrate their effectiveness and motivate the naming of the improved protocol as SHORT. Additionally, SHORT induces a more intense signal and better S/N ratio than the SWITCH classic version (Fig. 1c). As shown in figure S3c, the combined effect of $H_2O_2$ and AR enhances the labeling process with respect to the single treatments (either $H_2O_2$ or AR). At this point, we concluded that 37 °C was the best temperature condition for probing the staining efficiency. Treatments like SB, $NaBH_4$, and $CuSO_4$, which strongly reduced autofluorescence in SWITCH-processed slices (Fig. 1a, b and Fig. S1), induced a low-intensity peak in the first 50-100 μm and an emission signal comparable to the natural autofluorescence along with the whole tissue thickness at 638 nm (Fig. 1e, f). We also evaluated the $CuSO_4$ treatment before and after the immunostaining process, finding no improvement of the signal (Fig. S2e). This is attributable to the deposition of $Cu^{2+}$ ions which increases the scattering process and/or the epitope damaging of these treatments. While optimizing the primary and secondary concentrations of primary anti-NeuN and Alexa Fluor 647 antibodies, we demonstrated that the best dilutions are 1:50 and 1:500, respectively (Fig. S3b). On the other hand, SHIELD-processed slices showed an evident probe accumulation in the first 100 μm at different temperatures (4 °C and 37 °C) (Fig. 1d). Such results were also demonstrated by combining SHIELD with other autofluorescence elimination reagents and unmasking techniques (Fig. S2a, b, d). Finally, both the SHIELD and SWITCH protocol did not show efficient labeling in HEPES buffer, HEPES buffer supplemented with Quadrol and Urea, and PBST supplemented with BSA (Fig. S2b, d). The overall assessment of the tissue-processing procedures resulted in the optimization of SHORT (autofluorescence characteristics in Fig. S4): a combination of the SWITCH technique with $H_2O_2$/AR treatments, high-temperature incubation of antibodies, and TDE refractive index matching, to enable homogeneous labelling of human brain samples.

Figure 2a, b show the complete pipeline for cutting, processing, and acquiring 500-μm thick human brain slices. Briefly, SHORT is based on the SWITCH chemical principles to generate meshgel polymers from inside the tissue, to provide structural and biomolecular preservation (Fig. 2). After the glutaraldehyde fixation, lipids (responsible for preventing the staining to deep structures and inducing the scattering process during imaging) can then be removed without losing native tissue components using a clearing solution (200 mM SDS, 10 mM lithium hydroxide, 40 mM boric acid) at 55 °C (Fig. 2a). Next, SHORT requires the incubation in 30% vol/vol in $H_2O_2$ for 1 h at RT, and Tris-EDTA for 10 min at 95 °C (Fig. 2b). The resulting treatments show an efficient staining for different biological features by incubating the antibodies at 37 °C. After staining, the tissue-gel hybrid is immersed in a refractive index matching (TDE/PBS) and enclosed in a sandwich holder to stabilize and preserve the specimens (Fig. 2a). Also, a multi-round of staining, stripping, and acquisition on SHORT-process specimens can be performed for proteome investigation (Fig. 2a, b) as presented in the following paragraph.

**SHORT allows multicolor and multiround staining of the human brain tissues**. To perform multiple molecular phenotyping of tissues, we validated SHORT with various markers (high-resolution images in Fig. S5; Supplementary Movie 1–3), demonstrating its compatibility also with low-density proteins

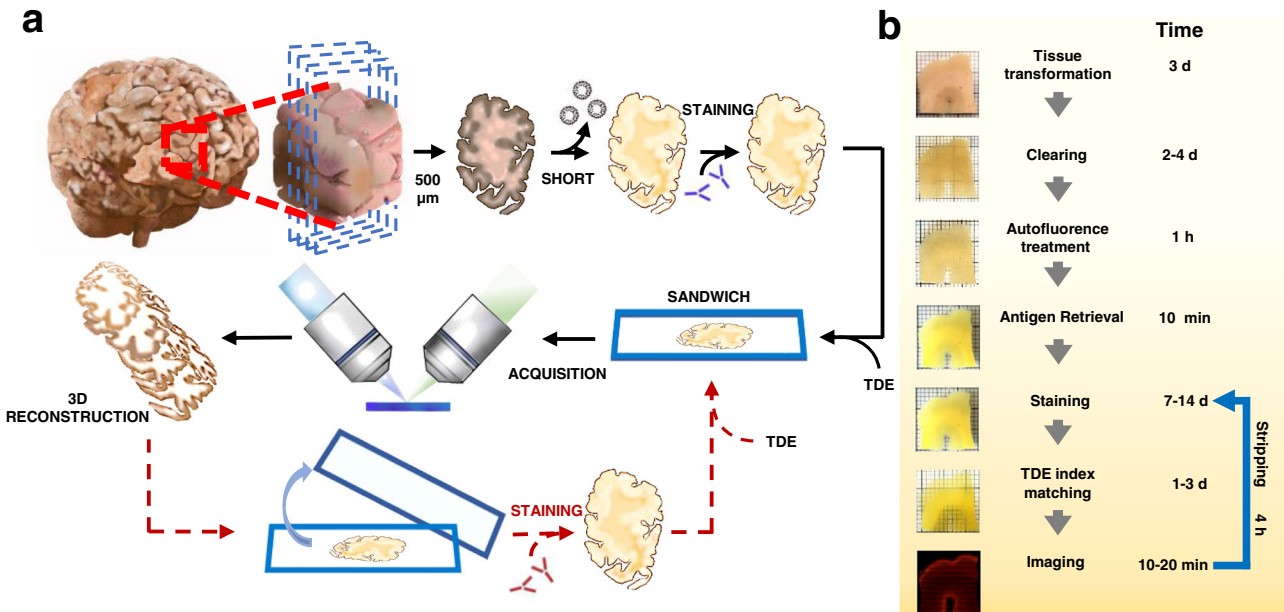

**Fig. 2 Outline and timing of SHORT. a** The entire SHORT protocol, from the whole human brain sample to imaging, requires three weeks. This consists mainly of incubation time, reflected in the 'Time' column for each step (**b**). During days 1-3, SHORT processing slices were performed. On days 4-8, the removal of the lipid takes place. On day 9, peroxide oxygen treatment (1 h) and AR (10 min) were performed. On days 9–15, the primary antibodies were incubated. The secondary antibody incubation was performed during days 15-19. Finally, the samples were acquired using LSFM and, then, reconstructed using Zstitcher. Additionally, the processed samples can undergo multiround staining for proteome analysis.

like parvalbumin (PV) and calretinin (CR), indeed we observed that the S/B (signal/background) measured overall the 500 μm depth of the sample, is sufficient to detect the labelled neurons (Fig. 3a). We verified the compatibility of SHORT with different nuclear stains, including Sytox Green, DAPI, and Propidium Iodide (Fig. S6; Fig. 4k). Also, to verify the compatibility of SHORT with lipophilic dyes, we tested the DiD staining and we visualized the myelinated fiber pathways in a 100-μm human superior frontal cortex slice (Fig. S7). Then, we optimized different strategies to co-stain the tissue up to three different markers and perform multiple rounds of labelling. To avoid cross-recognition between the antibodies we selected different species both for the primary and the secondary antibodies. In particular, we developed costaining protocols for: NeuN and glutamic acid decarboxylase (GAD67) (Fig. 4f and Fig. S8); NeuN and CR; NeuN and somatostatin (SST) (Fig. 3b); vasoactive intestinal peptide (VIP) with SST and PV; NeuN with CR and SST; glial fibrillary acidic protein (GFAP) with SST and vimentin (VIM) (Fig. 3c). All the antibodies and dyes used in the study are summarized in supplementary Table S1.

To access the phenotyping analysis potential of SHORT, we generated multi-round labelling of the tissue by removing the antibodies and restained the sample with other markers. To remove the staining completely, the antibodies were stripped off from the tissue by soaking the stained sample in the clearing buffer at 80 °C for 4 h. We observed that the stripping process reduces the autofluorescence signal of lipopigments (i.e., lipofuscin) by about 2 times (Fig. S9). We were able to perform three consecutive rounds of staining in 500 μm-thick slices of the human brain cortex, confirming effective antibody removal, preservation of neuronal morphology as well as re-staining capability. In particular, we first costained the tissue with anti-VIP, anti-SST, and anti-PV antibodies. We then stripped out the antibodies and we restained the tissue and repeated the imaging process twice, labelling for the second and third round, respectively: SST, CR, NeuN and SST, GFAP, VIM, using SST as a fiducial marker (Fig. 3c). We achieved multiplexed profiling

of 7 different targets in the same piece of human superior frontal cortex by multispectral imaging of 3 antigens in each round. To evaluate if the stripping process affects the staining of the tissue, we quantified the signal-to-noise ratio (SNR) of SST immunostaining at rounds 1, 2 and 3, (N = 30 SST-immunoreactive neurons for each round), finding a loss of 30% protein signal during the first round (Fig. 3d). Our data shows that the SNR of SST immunostaining in rounds 1, 2, and 3 decreases from round 1 to round 2. This suggests that the amount of protein lost in the stripping process decreases after every round, but antigens are still preserved and they can be detected using SHORT. To verify the absence of distortion associated with the high-temperature incubation during the stripping process at cellular level, we quantified the structural similarity index (SSIM index[54]) of 3 different SST-immunoreactive neurons and their dendritic branches tortuosity (n = 6) before and after stripping. High SSIM index value (0.9 ± 0.06) (Fig. S10a) was found and no significant structural change in the branching patterns (Fig. S10b) was observed.

**Volumetric 3D reconstruction of SHORT cleared samples with light-sheet fluorescence microscopy.** The investigation of the 3D molecular tissue phenotyping of large human brain samples was performed by coupling SHORT with a custom-made LSFM that enables imaging of large samples up to 30 × 30 cm² thanks to a large sample accommodation chamber (see Methods for further details). Tissue blocks of the hippocampus, precentral gyrus, prefrontal cortex, motor cortex, and Broca's area (Table S2) of approximately 3 cm³ in size were cut with a vibratome (Compresstome VF-900-0Z) into slices up to 500 μm of thickness and treated with SHORT for tissue clearing and labelling. Samples were mounted in a custom-made sealed sample holder (termed "sandwich", Fig. S11) suitable for both imaging and long-term storage. Indeed, we observed a preservation of fluorescence signals of SHORT-processed slices stored in the sandwich for 1 year. We compared the signal obtained from a first acquisition of a

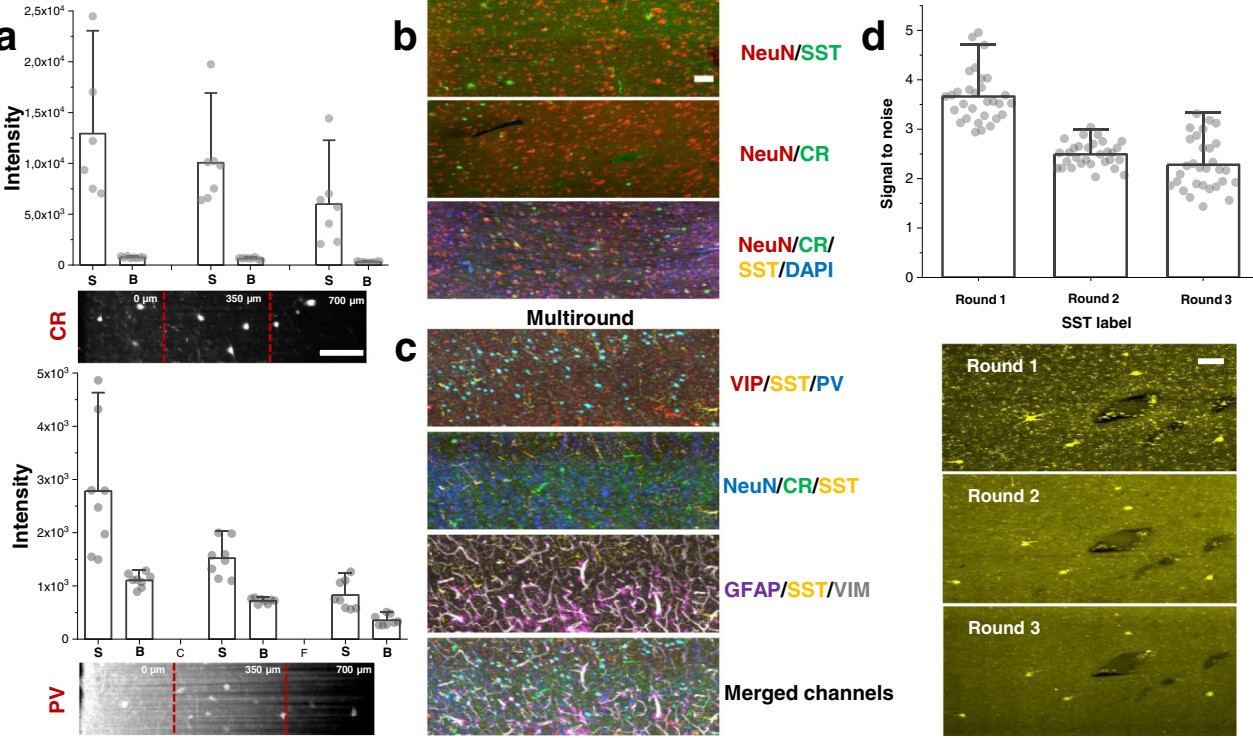

**Fig. 3 Multiplexed staining and antigen preservation of SHORT-processed tissue. a** Labelling efficiency of two neuron markers, PV and CR, acquired by LSFM. Mean and SD were calculated by taking the signal of 8 different neurons and background in three different regions: 0–240 μm corresponding to the maximal light exposure (0 μm), 240–470 μm corresponding to the middle of the slices (350 μm), and 490–700 μm which was opposite to the excitation side (700 μm). Abbreviations: S, signal; B, background. High-resolution images, scale bar = 100 μm. **b** Different examples of optimized costaining in SHORT processed slices. Maximum intensity projection (MIP) of 20 slices. Downscaled images, scale bar = 100 μm. **c** Three distinct rounds of immunostaining for probing neurons (NeuN), inhibitory markers (Vip, SST, PV, CR) vasculature (VIM) and glial cells (GFAP). MIP of 70 slices. Downscaled images neuron, scale bar = 100 μm. **d** SNR was quantified for SST immunolabeling in round 1, round 2, and round 3 (n = 30 neurons). MIP of 70 slices. Downscaled images, scale bar = 100 μm.

Broca Area labeled with CR, SST, and NeuN, with the one acquired after one year, detecting a reduction of 51% for Alexa Fluor 488; 24% for Alexa Fluor 568; and 14% for Alexa Fluor 647 (Fig. S12) (no photobleaching process due to re-acquisition was detected). In addition to RI matching, TDE also leads to tissue dehydration, which reduces slightly the specimen volume (Fig. S13) and preserves the tissue, a highly beneficial effect for routine acquisition and long storage of labeled specimens. The 3D reconstruction of human cortical cytoarchitecture was performed using the high-throughput capability of our LSFM which, employing an isotropic resolution of $3.3 \times 3.3 \times 3.3 \ \mu m^3$, enables to resolve single neuronal subtypes and soma morphology features of their individual subpopulations (Fig. 4b–f). The clearing process was highly effective in the gray and white matter (Fig. 4a) and the antibody labeling was homogenous along with the entire depth of the samples (Fig. 4f). Although we obtained a good probe penetration depth at different excitation wavelengths (488, 561, and 638 nm) (Fig. S12 and S14; Supplementary Movie 4), we encountered difficulties in achieving a similar degree of staining homogeneity and light penetration depth at 405 nm with the nuclear staining DAPI. This effect is known and termed the inner filter effect[55] in the literature and will require further work to be overcome.

We first used NeuN combined with GAD67 to perform multiple labelling throughout the $1.5 \times 1.5 \times 0.5 \ cm^3$ hippocampus human brain slices (Fig. 4b–f). Figure 4d, e show the feasibility of such methodology by imaging hippocampal neurons of the CA1 as well as the human dentate gyrus SHORT-processed slices. Furthermore, the multiple staining approach was also

performed on human brain slices from Broca's area (Fig. 4g–j), and prefrontal cortex (Fig. 4k–n), using three different antibodies against different neuron types, including SST, CR, as well as NeuN. It is known that CR- and SST-immunoreactive interneurons were distributed in all cortical layers[56], but there is little information in the literature on the distribution of SST-expressing neurons in white matter. Our measurements show that white matter regions are rich in SST-expressing neurons (Fig. 4o) in the prefrontal cortex.

In addition, the multi-round characterization enabled by SHORT allowed us to obtain homogeneous labeling of seven different markers, including all neurons (NeuN), inhibitory neuron markers (SST, CR, PV, VIP), glial cells (GFAP), and vasculature (vimentin, VIM), as demonstrated by the successful imaging of a slice from the inferior frontal pole (Fig. 5a, b). Thanks to the specific mounting in the sandwich holder (Fig. S11), the tissue orientation could be maintained through the acquisitions, allowing a good coregistration of the different acquisitions and facilitating future quantitative interrogation of the data.

## Discussion
Although several brain tissue-clearing and labeling methods are available, few of them were originally optimized on the human cortex. Indeed, prolonged formalin fixation and variable post-mortem conditions—such as pH variation and tissue oxygenation—could affect the quality of immunostaining due to the epitope masking effect and antigen damage. Using as a starting point the SWITCH/TDE method and coupling it with conventional buffers

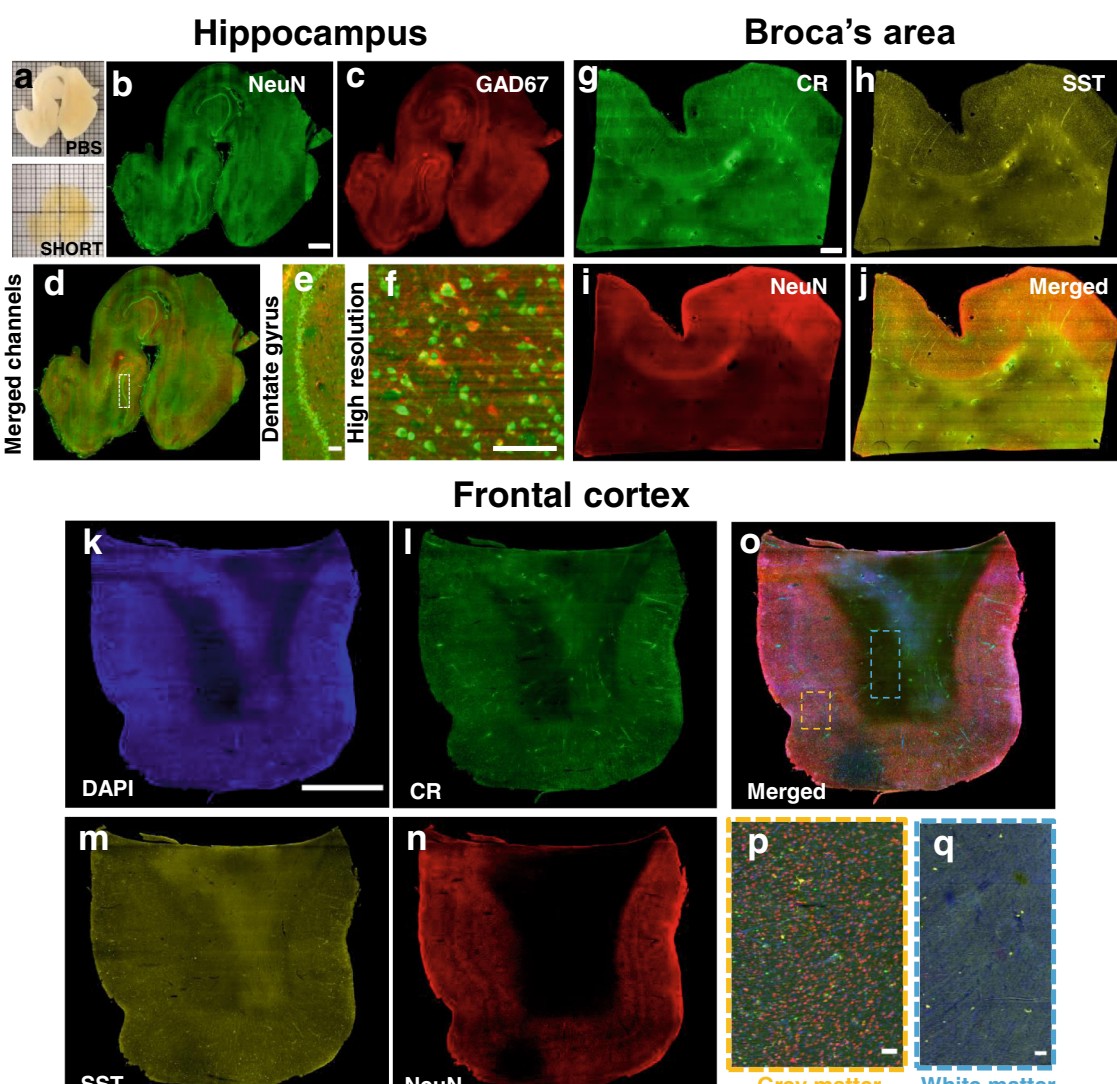

**Fig. 4 SHORT combined with LSFM enables single cell-resolved imaging of large portions of adult human slices. a** Image of a typical 500 µm-thick adult hippocampus slice before and after SHORT. **b** Maximum intensity projection (number slices: 140) image shows a mesoscopic reconstruction of the processed hippocampus slice labeled for NeuN (excitation light 488 nm) and (**c**) GAD67 (excitation light 561 nm), and (**d**) merged channels. Scale bar = 1 mm. **e** Higher magnification on the dentate gyrus and (**f**) example of high-resolution single cell resolved imaging with our custom-made LSFM. Scale bar = 100 µm. (**g–i**) MIP of 20 slices of Broca's area labeled for CR (**g**), SST (**h**), and NeuN (**i**). **j** Merged channels. Scale bar = 1 mm. **k–q** MIP images (slice number 20) of the frontal cortex labeled using the nuclear marker DAPI (**k**), CR (**l**), SST (**m**), and NeuN (**n**). **o** Merged channels. Scale bar = 4 mm. **p** Magnified images of the white matter (blue rectangle), characterized by a high distribution of SST-immunoreactive neurons, and (**q**) grey matter (orange rectangle). Scale bar = 100 µm.

used in classical immunocytochemistry, we developed and optimized SHORT, an efficient clearing and labelling method for 3D human cortex molecular probing with LSFM. In general, SWITCH allows preserving the endogenous biomolecules, while TDE performs sample clearing. However, we encountered several hurdles for performing volumetric reconstruction of the human cortex. The first challenge is the long fixation time, which could cause a variety of changes in the 3D structure of proteins, decreasing the possibility to detect specific epitopes for phenotyping and connectomes analyses[57,58]. Also, the presence of lipopigments—such as lipofuscin and neuromelin[59–61]—and blood, have remained an insuperable hurdle during the acquisition process, causing spurious signals that make cell identification and counting more complex. In addition, aldehyde fixation using formaldehyde and glutaraldehyde leads to a wide emission spectrum resulting in high background fluorescence that limits the achievable SNR. Indeed, the generation of the double bonds

between the crosslinker and proteins increases the autofluorescence, making the sample unsuitable for immunofluorescence assay. To overcome this limitation, SHORT was engineered to be capable of decolorizing such autofluorescence, blood-infused tissue and lipopigments, in combination with epitope restoration. This strategy allowed to improve the image quality, the antibody's penetration, and the SNR of reconstructed large portions of human brain slices. Also, we investigated several autofluorescence elimination reagents commonly used in conventional immunocytochemistry. Some of them, such as SB[45,46], NaBH$_4$[3,43,44], and CuSO$_4$[43,45,46], significantly reduced the autofluorescence signal, but they were not compatible with our method, because they increase the scattering process during the acquisition by forming stable complexes (such as Cu$^{2+}$ and lipofuscin), or causing epitope damage. For these reasons, the best option for tissue structural preservation and autofluorescence quenching capability was provided by peroxide hydrogen,

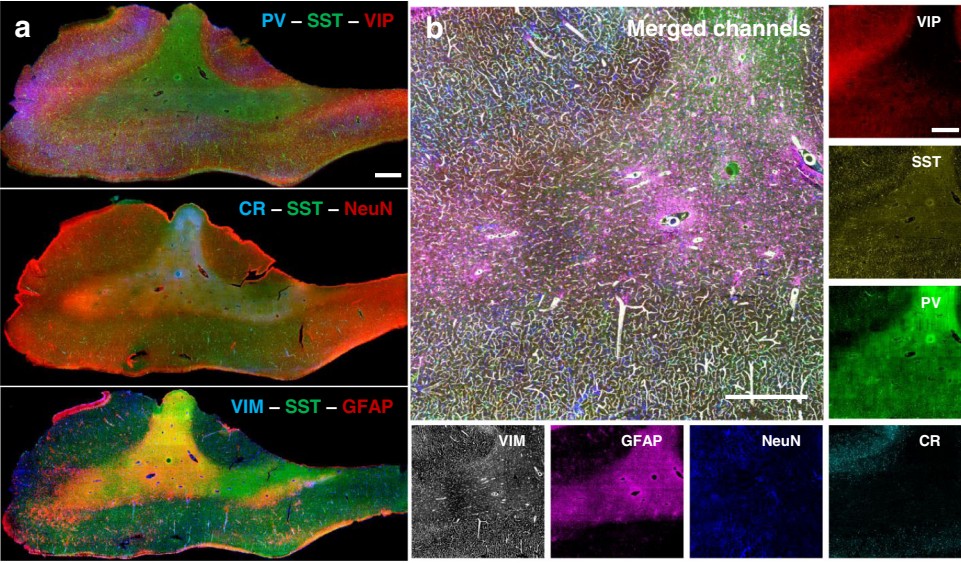

**Fig. 5 SHORT allows multiple rounds of staining, imaging, and stripping of large portions of human brain slices. a** Seven different antibodies were used across 3 sequential rounds of immunostaining, with SST employed as fiducial markers in each round. Round 1: PV—SST—VIP; round 2: CR- SST—NeuN; round 3: VIM—SST—GFAP. Scale bar = 1 mm. **b** The merged image shows a particular of the white and grey matters labeled with VIP, SST, PV, CR, NeuN, GFP, VIM. MIP of 70 slices. Scale bar = 100 μm.

coupled with the unmasking effect of antigen retrieval, a combination that suggested naming our method SHORT (SWITCH—$H_2O_2$—antigen Retrieval—TDE). We established that our approach allows staining 500-μm-thick human slices with different ages and fixation times and performing volumetric acquisition using LSFM. Although LSFM lacks axial resolution power, it is the best optical imaging technique for probing multiple markers in large specimens. Indeed, LSFM offers fast acquisition time and low photobleaching with respect to other conventional optical techniques, such as confocal and two-photon fluorescence microscopy[2]. The probe penetration efficiency with antibodies against a variety of antigens, including NeuN, SST, and CR, which could be masked following SDS treatment[24], was investigated by LSFM. Although SHIELD[25] was fully applied to clear formalin-fixed 2-mm-thick coronal blocks of the human brain (9 × 5.5 × 0.2 cm³) and label with different markers, our replication of this approach showed a not homogeneous labelling for high density-epitopes. This is likely attributable to the dense epoxy meshgel and antibody choice, which can induce a probe accumulation in the first 100 μm of the slice. With SHORT, volumetric blocks of the human brain can be reconstructed by cutting the area of interest in slices of 500 μm of thickness. In this paper, we showed the reconstruction of 500 μm-thick slices of various areas of the brain demonstrating the versatility of the method. In addition, SHORT was applied in Costantini et al.[62] demonstrating how this approach can be used to reconstruct a portion of Broca's area of 1.5 × 1.3 × 0.8 cm³ cutting it in only 16 slices and allowing the cell census of this volumetric portion of the cortex. The entire dataset is available on the DANDI archive (https://gui.dandiarchive.org/#/dandiset/000026).

Exploiting the structural and epitope preservation of such methods, we then attempted and successfully performed multi-rounds of immunostaining and stripping in the SHORT-processed human cortex. Indeed, optical microscopy is limited by the visible light bandwidth and the host of animal species in the classic immunocytochemistry, which makes the detection of more than four different markers challenging without resorting to complex multiplexing techniques that require computational spectral disentangling[63–65]. The recent advantages of the multiplexed tissue strategies, including MAP[29,66], SWITCH[23],

SHIELD[25], and miriEx[67] have allowed bypassing this limit. These works demonstrated that specific antigens can be repeatedly labeled without loss of antigenicity in 100-μm thick human and mouse brain samples[23,67]. In agreement with these results, we optimized the possibility to perform multi-rounds in SHORT-processed 500-μm-thick human brain slice, by labeling neuronal (VIP, SST, NeuN, PV), glial (GFAP), and vasculature markers (VIM) in three distinct rounds of immunostaining and stripping. The simultaneous acquisition of three different markers for each round makes easier the co-registration of different channels in a large portion of the human cortex. We showed robust preservation of epitopes following clearing, antigen retrieval, and stripping cycles in a processed large human portion.

We also performed staining using DiD on 100 μm-thick samples of superior frontal cortex, demonstrating the compatibility of SHORT with lipophilic tracing for the rapid visualization of human cytoarchitecture using other methods than immunohistochemistry. In future, the advantages of SHORT may allow combining molecular details obtained with antibody staining and cellular connectivity tranced with lipophilic dyes from diverse cell types within the same tissue.

Although SHORT allows simultaneous detection of multiple markers and multi-round cycles without compromising the tissue integrity, such protocol still presents potential limitations. In particular, the decreased penetration efficiency at 405 nm for nuclear stains like DAPI and the substantial attenuation of the excitation beam in such samples, termed the inner filter effect[55], should also be improved in the future. To facilitate sample mounting and obtaining human slices with a flat surface we developed a sample "sandwich" holder that also permits long-term storage of the specimens. Although our pipeline allows modulating the XY dimension of the samples using appropriate sandwiches that are size-adjustable, we are still developing specific software for aligning and stitching non-rigidly subsequent the human cortex slices imaged with the LSFM to obtain consolidated 3D maps and reconstructions of the sample's volume.

In conclusion, we presented a new method, named SHORT, able to preserve the tissue integrity with epitope restoring in centimeter-sized human brain specimens for molecular and cellular phenotyping. Such methodology allowed homogenous tissue

labeling of different markers and clearing of human specimens. Also, we implemented a new pipeline compatible for long-term tissue storage of such precious samples. Thus, in combination with fast light-sheet microscopy systems, SHORT could be a key technology to map large human brain regions, up to the whole organ, accelerating the understanding of its structural and cellular characteristics, as well as of physiological and pathological conditions underlying such a complex system.

## Methods

**Samples**. Brain tissue specimens were collected from 3 different human donors with no known neuropsychiatric disorders that were obtained from the body donation program of the Université de Tours and from the Massachusetts General Hospital (MGH). The tissue donors provided their informed and written consent to the donation of their body for research purposes. We used 5 different cortical regions from these donors: the precentral cortex from donor 1 was used for autofluorescence analysis, the NeuN depth analysis, and stripping efficiency validation; hippocampus from donor 1, prefrontal cortex from donor 2, motor cortex, and Broca's area from donor 3 were used for antibody validation, stripping process, and volumetric acquisitions (Table S2). All cases were neurotypical and had no neuropathological diagnosis of neurodegenerative disorder. Tissue sections of $450 \pm 50\,\mu m$-thick coronal sections were obtained using a custom-made vibratome (Compresstome VF-900-0Z). The slices were stored at 4 °C in PBS 0.02% NaN$_3$. For preserving the tissue architecture in such a sample, SHORT and SHIELD protocols were used. The common idea of these methods is to diffuse and then crosslink fixative molecules (glutaraldehyde or polyglycerol-3-polyglycidyl) throughout the samples, by changing the buffer characteristics. Then, the transformed tissue is treated with strong detergent [sodium dodecyl sulfate (SDS)] for lipid removal. During the procedure, in particular after the clearing process, adequate care for the fragile samples was performed.

**SHORT protocol for human brain slices**. An adapted version of the SWITCH/TDE protocol from Costantini et al.[6] was used to clear large portions of the human brain samples. The specimens were incubated in a Switch-Off solution, consisting of 50% PBS titrated to pH 3 using HCl, 25% 0.1 M HCl, 25% 0.1 M potassium hydrogen phthalate (KHP) and 4% glutaraldehyde. After 24 h, the solution was replaced with the SWITCH-On solution, containing PBS pH 7.4 with 1% glutaraldehyde. After 3 washes for 2 h each in PBS at RT, the specimens were inactivated by overnight incubation in a solution consisting of 4% glycine and 4% acetamide at 37 °C. Following inactivation, the samples were extensively washed in PBS and then incubated in the clearing solution containing 200 mM SDS, 10 mM lithium hydroxide, 40 mM boric acid for 2-4 d at 55 °C depending on the sample size. After the clearing process, the samples were washed 3 times in PBS + 0.1% Triton X-100 (PBST) at 37 °C for 24 h. Next, the slices were treated with 30% H$_2$O$_2$ for 1 h and, after 3 washes for 10 min each, antigens were retrieved with the preheated tris-EDTA buffer (10 mM Tris base (v/v), 1 mM EDTA solution (w/v), 0.05% Tween 20 (v/v), pH 9) for 10 min at 95 °C in a volume of 20 ml. This step allows increasing the sensitivity of reaction of antibodies directed to specific targets. For reaching a good success rate, the cleared samples must swell by about 1.5 times. After cooling down to RT, the specimens were washed in DI water for 5 min each and then equilibrated with PBS for 1 h.

**SHIELD/TDE protocol for human brain slices**. The reagents for SHIELD[25] were provided by LifeCanvas. Briefly, the tissue slices were incubated for 24 h at 4 °C with shaking for 1 day in the SHIELD-Off solution (10 ml SHIELD-Epoxy Solution, 5 ml SHIELD-Buffer Solution, 5 ml DI water). Next, the samples were incubated at RT with shaking for 1 day using SHIELD-On Buffer and SHIELD-Epoxy Solution mixed in a ratio of 1:1 (final volume of 40 ml). Both SHIELD-Off and SHIELD-On were freshly prepared and not used after two months from the opening. Finally, the samples were cleared for 3 days at 55 °C using 200 mM SDS, 10 mM lithium hydroxide, 40 mM boric acid and washed 3 times in PBST at 37 °C for 24 h.

**Alternative decolorization treatments (autofluorescence analysis)**. After clearing, small portions of the precentral cortex (99-year-old (female), 6 months in formalin, donor 1, Table S2) were used for investigating the optimal autofluorescence masking treatments. Such treatments were probed on SWITCH/SHIELD-processed tissue slices ($N = 4$ independent experiments for each autofluorescence masking treatment were performed in the precentral cortex, donor 1). To promote and enhance their effects, some of these reagents were combined together.

- Oxygen peroxide: the tissue slices were soaked in 0.6%, 15%, and 30% (v/v) H$_2$O$_2$ diluted in DI water for 1 h at RT. Sections were then washed with PBS 3 times, for 10 min each.
- Sodium borohydride: the tissue slices were immersed in 0.1%, 1% and 2% (w/v) NaBH$_4$ in DI water for 1 h at RT. Next, the samples were washed with PBS 3 times, for 10 min each.

- Tris-EDTA Buffer Antigen Retrieval: antigens were retrieved in pre-heat tris-EDTA buffer consisting of 10 mM Tris base (v/v), 1 mM EDTA solution (w/v), 0.05% Tween 20 (v/v), pH 9 for 10 min at 95 °C. Next, the samples were cooled at RT for 30 min and washed in DI water 5 min each. Finally, the slices were equilibrated in PBS for at least 1 h.
- Autofluorescence Elimination Reagent (SB): this reagent was applied before and after the immunostaining protocol. After extensively washing in PBS, the sections were immersed in 70% ethanol for 5 min. Next, the ethanol was replaced with SB for 5 min. Afterwards, 3 changes of 70% ethanol for 1 min each were performed. Finally, the samples were equilibrated in PBS.
- Ascorbic acid: processed slices were treated with 6 mM ascorbic acid for 1 h and washed 3 times in PBS for 5 min each.
- Hydroquinone: sections were immersed in an aqueous solution containing 20 mM Hydroquinone for 1 h at RT and washed 3 times with PBS for 5 min each.
- Quadrol: sections were incubated for 2-4 h in 25% Quadrol and extensively washed with PBST.
- Copper (II) Sulfate (CuSO$_4$): this reagent was applied before and after the immunostaining protocol. Sections were treated with 1 mM and 10 mM CuSO$_4$ in 50 mM ammonium acetate for 1 h, then washed with PBST for 3 times.

For all the treatments, the sample tests were then equilibrated with 30% TDE/PBS (v/v) for 1 h and 68% TDE/PBS (v/v) for, at least, 1 h, before the placement in the sandwich and acquired using our LSFM.

**Immunolabeling protocol**. After clearing, small portions of the precentral cortex (99-year-old, female, 6 months in formalin, Table S2) were used for investigating the optimal probe penetration using different autofluorescence masking treatments, temperatures, and dilution buffers ($N = 3$ independent experiments were performed for each treatment, temperatures, and dilution buffer in the precentral cortex, donor 1). For the optimization of the immunolabeling protocols, the antibody incubations were performed at 37 °C and 4 °C, using the following buffers: PBST, HEPES buffer 1X (HEPES 10 mM pH 7.5, 10% (v/v) Triton X-100, 200 mM NaCl, stock solution 2X), HEPES buffer supplemented with 2.5% Quadrol and 0.5 M urea. For the NeuN depth test, the primary antibody was incubated for 3 d at the dilution 1:50 and 1:100. Afterwards, the anti-NeuN stained samples were incubated for 1 day with the secondary antibody (dilution 1:200, 1:500, 1:1000). These protocols were performed for all the autofluorescence masking treatments described previously ($n = 3$) using the precentral cortex (99-year-old (female), 6 months in formalin).

For the double and triple staining, the primary antibodies (anti-NeuN, anti-SST, anti-CR diluted at 1:50, 1:200 and 1:200, respectively) were incubated for 7 days at 37 °C. After 3 washes for 30 min each at 37 °C with PBST, the stained sample was incubated with the secondary antibodies alpaca-antirabbit IgG AF 488, donkey anti-rat IgG AF 568, goat antichicken IgY AF 647 (see Table S1), for 5 days at 37 °C and, then, washed 3 times for 1 h each at 37 °C. Such optimized staining protocol was performed on the superior frontal cortex (Figs. 3 and 4k–q; donor 2), Broca's area (Fig. 4g–j; donor 3). For the hippocampus slice (Fi. 4b-f; donor 1), the primary antibodies (anti-NeuN, anti-GAD67 diluted at 1:50, 1:200, respectively) were incubated for 7 days at 37 °C. After 3 washes for 30 min each at 37 °C with PBST, the stained sample was incubated with the secondary antibodies goat-anti chicken IgG 488 and donkey anti-mouse IgG 647 (see Table S1), for 5 days at 37 °C and, then, washed 3 times for 1 h each at 37 °C.

For the multi-round experiments, the tissue was processed and stained as described previously. After acquisition, the samples were removed from the sandwich holder, equilibrated in 30% TDE/PBS (v/v) for 1 h and washed 3 times in PBS for 1 each. Then, the specimens were delabeled with the clearing solution (200 mM SDS, 10 mM lithium hydroxide, 40 mM boric acid) at 80 °C for 4 h. Subsequently, the samples were extensively washed with PBST for 24 h. The stained proteins were: PV, SST, and Vip, labeled with bovine anti-goat IgG AF 488, donkey anti-rat IgG AF 568, and alpaca anti-rabbit IgG AF 647, respectively. After delabeling, CR, SST, and NeuN were stained with alpaca-antirabbit IgG AF 488, donkey antirat IgG AF 568, goat antichicken IgY AF 647 (second round). Finally, VIM, SST, and GFAP were labeled with goat-anti mouse IgG AF 488, donkey anti-rat IgG AF 568, and alpaca-anti rabbit IgG AF 647 (third round). Next, the stained samples were equilibrated with 30% TDE/PBS (v/v) and 68% TDE/PBS (v/v) supplemented with DAPI diluted 1:100 for 1 day. Afterwards, the samples were placed in the sandwich with 68% TDE supplemented with DAPI at the dilution of 1:2000.

**Lipophilic dye staining with DiD**. Slices (100 μm-thick) of superior frontal cortex from donor 2 were processed with SHORT and then incubated in 0.5% DiD (1,1'-dioctadecyl-3,3,3',3'-tetramethylindodicarbocyanine, 4-chlorobenzenesulfonate salt) (stock solution 1 mg/ml, dilution 1:2) dissolved in 10% SDS (stock solution 20%, dilution 1:2) at 37 °C for 2-3 days. Samples were then soaked in a solution of 0.5% DiD in PBST and incubated at 37 °C with gentle shaking for 1 day. Finally, the samples were incubated with 30% TDE/PBS (v/v) and 68% TDE/PBS (v/v) in gentle shaking to remove the excess of DiD and match the refractive index.

**Imaging**. The tissue sections were imaged with our custom-made inverted light-sheet fluorescence microscope[68]. The illumination and detection objectives are a pair of LaVision Biotec LVMI-Fluor 12x PLAN whose main characteristics are 12x magnification, NA 0.53, WD 10 mm, with a correction collar for refractive index matching with the immersion solution. The objectives are orthogonal and tilted at 45° in respect to the sandwich holder plane to avoid placing mechanical constraints on the lateral extension of the samples. Additionally, the set-up includes 4 over-lapped laser lines (Cobolt MLD 405, MLD 488, DPL 561 and MLD 633 nm) with the corresponding fluorescence filter bands. An acousto-optical-tunable-filter (AAOptoelectronic AOTFnC-400.650-TN) modulates the transmitted power and wavelength and a galvo mirror (Cambridge Technology 6220H) sweeps it across the detection focal plane, generating a digitally scanned light sheet. The induced fluorescence is collected by the objective onto a Hamamatsu ORCA Flash4.0v3 sCMOS camera, operating in the confocal detection mode[4,69]. Field of view of the camera: 2048 × 2048 pixels, with a pixel size of $0.55 × 0.55 × 3.3 \, \mu m^3$. The sample, enclosed in a sandwich holder, is accommodated in a large plexiglass chamber filled with 68% TDE or 91% glycerol and into which the objectives are immersed in order to allow for refractive index matching. Both the spacer, the holding glass and the cover glass are purchased from companies (Microlaser srl, Sesto Fiorentino, Italy; Laser Optex Inc., Beijing, China) that produce customizable size items permitting to adjust the size of the sandwich according to the application. The sample is imaged by translating it along the horizontal direction in a snake-like pattern, using a precision three-axis translation stage system (two PI M-531.DDG and a PI L-310.2ASD for a maximum motion range of $30 × 30 × 2.5 \, cm^3$). The whole experimental apparatus is controlled using a dedicated data acquisition and control software written in C++, which was specifically developed for this setup to be able to sustain data rates as high as 1 GB/s.

For stripping validation, unstained, stained for NeuN antibodies with Alexa Fluor 568, and stripped precentral slices were acquired using a Nikon C2 laser-scanning confocal microscope with a Plan Fluor 60×/1.49 NA oil immersion objective at 561 nm. Laser power 2 mW.

**Statistics and Reproducibility**. For tissue autofluorescence characterization, cleared samples were imaged at 405, 488, 561, and 638 nm using our custom-made LSFM. To quantify the fluorescence intensities in each channel, 20 regions of interest (ROI) of $20 × 500 \, \mu m^2$ were selected randomly in the grey matter of volumetric reconstruction of $≈5 × 5 × 0.5 \, mm^3$, using Fiji[70]. The 20 ROIs were analyzed for each treatment and tissue transformation protocol on consecutive small pieces of cortex (precentral cortex) from patient "donor 1" (Table S2). We decided to characterize the effect of the various reagents on the brain of a very old patient (99 years old) as it presented the more severe presence of lipofuscin, whose accumulation increases during aging leading to an enhancement of the auto-fluorescence signals. The laser power at 405, 488, 561, and 638 nm was set at 5 mW. Means and standard deviations were plotted using Origin (OriginLab Corporation, Origin 2019b). The significance of the difference in mean values was determined using the Mann-Whitney test (Origin 2019b). The statistical significance was set at an α level = 0.05 (*$P < 0.05$; **$P < 0.01$; ***$P < 0.001$; ****$P < 0.0001$). To guarantee the reproducibility of the findings, 4 independent experiments were performed on different slices. For the NeuN depth analysis, treated SWITCH/SHIELD processed slices were acquired using our LSFM at 638 nm and $n = 3$ of $200 × 700 \, \mu m^2$ ROIs were selected for each treatment by Fiji, and mean and standard deviation were plotted using Origin. For quantifying the antibodies removal, ROIs of $6 × 6 \, \mu m^2$ of images acquired at 561 nm using confocal microscopy were selected by Fiji, their mean and SD were plotted using Origin. For the shrinking quantification, the sample area before and after the clearing process was calculated by Fiji ($N = 3$).

For signal retention, 10 neurons CR-, SST-, and NeuN-immunoreactive in a region of $500 × 500 \, \mu m^2$ were selected in the following downsampled reconstructions: after the immunostaining process (column 1; mean and standard deviation (SD)), after 1 year in the sandwich holder (column 2; mean and SD), and then reacquired after another week for the quantification of the bleaching process (3° column; mean and SD; see Fig. S12). The signal retention was calculated as signal to the background by taking 10 ROIs corresponding to the cellular soma ($10 × 10 \, \mu m^2$) and 10 ROIs of the background ($10 × 10 \, \mu m^2$) using Fiji, and plotted with Origin.

The degree of the deformation of the tissue caused by SHORT during the stripping process was performed on the high-resolution LSFM images by using the structural similarity index (SSIM index[54]) and the morphological deformation of the branching patterns at single-cell level in SST-immunoreactive neurons. For the SSIM index, 3 different pairs of neurons were cropped and scaled up by a factor of 4 to obtain a minimum number of $256 × 256$ pixels. Then, the images were processed using the plugin SSIM index, to quantify the structural similarity.

For the branching pattern tortuosity, the minimum distance and the real branch length of 6 different dendrites (see Fig. S10) was calculated using Fiji and plotted with Origin.

For visualization of the acquired data, the tiled raw images of extended samples acquired in a snake-like pattern by our LSFM are stitched into a 3D dataset. First, an affine transformation is applied to each image stack to resample the images and convert them from the objective reference frame to the sample reference frame. This is necessary to compensate for the fact that the objectives are tilted at 45°

while the sample moves horizontally, i.e. parallel to the ground, during the acquisition. This transformation effectively "re-slices" the image stacks, optionally applying a downsampling factor in a single transformation matrix. Afterimage reslicing, the resolution of the resulting images is isotropic at $3.3 \, \mu m^3$. Assuming Z is the thickness axis, X is the direction along which the sample moves and Y is the transversal axis, the resliced stacks are then stitched together along the Y axis using ZetaStitcher [https://github.com/lens-biophotonics/ZetaStitcher], a Python package designed to stitch large volumetric images such as those produced by LSFM. For visualization only purposes, a custom illumination intensity homogenization algorithm is applied to the stitched dataset or to its maximum intensity projection in order to equalize variations in the laser beam power that may occur within the objective field of view and/or between different image stacks. For each imaged wavelength independently, the algorithm simply averages the observed intensity along the X axis to create a smoothed intensity profile along Y that is used as a flat field reference to even out illumination intensity artifacts occurring across the transversal sample extension. After compression of the tiled raw images to JPEG2000 format and stitching the downscaled rows, the 3D downscaled rendering of the whole slices was performed by Fiji. Maximum Intensity Projection of 20-150 slices were used in Figs. 3, 4, 5, S6, S7, S12, and S14.

As example: the acquisition of a 500-μm thick, slice of a Broca's area of $2 × 2 \, cm^2$ stained with 2 different dyes requires 30 min and generates 0.4 terabyte. To efficiently store the 3D subvolumes (termed tiles) acquired by LSFM, the files were compressed to JPEG2000 format, which allows a reduction of the large-scale data by 60 times. To process and extract biological meaning from the acquired image data, the tiled raw images were resliced and downscaled with an isotropic resolution of $3.3 × 3.3 × 3.3 \, \mu m^3$ (starting resolution: $0.5 × 0.5 × 3.3 \, \mu m^3$). Next, the multi-tiles downsampled images were fused to perform a downscaled reconstruction of the whole slice. Such image processing allows converting the stained Broca's area of $2 × 2 \, cm^2$ in a Z-stack of 6 gigabyte (3 for each color) which can be visualized and analyzed by Fiji.

**Reporting summary**. Further information on research design is available in the Nature Research Reporting Summary linked to this article.

## Data availability
All data needed to evaluate the conclusions in the paper are present in the paper and/or the Supplementary Materials. Raw data other than the representative images are available from the corresponding author upon reasonable request.

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

## Acknowledgements

We thank Leah Morgan and Bruce Fischl, Massachusetts General Hospital, A.A. Martinos Center for Biomedical Imaging, Department of Radiology, USA for providing the human brain specimen 2 analyzed in this study and Jiarui Yang and David Boas, Boston University, Department of Biomedical Engineering, USA, for performing the slicing of the sample. We express our gratitude to the donor involved in the body donation program of the Association des dons du corps du Centre Ouest, Tours, and of the Massachusetts General Hospital who made this study possible by generously donating his body to science. This project has received funding from: European Union's Horizon 2020 research and innovation Framework Programme under grant agreement No. 654148 (Laserlab-Europe); European Union's Horizon 2020 Framework Programme for

Research and Innovation under the Specific Grant Agreement No. 785907 (Human Brain Project SGA2), No. 945539 (Human Brain Project SGA3) and under the Marie Skłodowska-Curie grant agreement No. 793849 (MesoBrainMicr); General Hospital Corporation Center of the National Institutes of Health under award number U01 MH117023; Italian Ministry for Education in the framework of Euro-Bioimaging Italian Node (ESFRI research infrastructure); "Fondazione CR Firenze" (private foundation). The content of this work is solely the responsibility of the authors and does not necessarily represent the official views of the National Institutes of Health.

## Author contributions

L.P., A.L. optimized the clearing and decolorization protocols; L.P., I.C., A.L. tested compatibility with immunolabeling. L.P., M.S. tested the different temperature for the antibody penetration and performed multi-staining and multi-rounds; M.S., L.P., and G.S. optimized the sample holder sandwich. L.P., V.G., and N.B. performed light-sheet imaging and confocal imaging. L.P., G.M., and V.G. contributed to the analysis and image processing. C.D. prepared the human brain slices. V.G., M.S., P.R.H, L.S. wrote parts of the manuscript. I.C., L.P., F.S.P conceived the study, supervised the study, analyzed results, and wrote the manuscript. All authors read and approved the manuscript.

## Competing interests

The authors declare no competing interests.
