## [Peer Review File · Communications Biology]

Reviewers' comments:

Reviewer #1 (Remarks to the Author):

In the present manuscript the authors describe an optimized clearing approach (SHORT) which is based on the earlier clearing protocol SWITCH. They apply this to thick sections of human cortical tissue and combine the clearing multicolor immunostaining and visualize the clearing and labelling results using light sheet fluorescence microscopy.

At first glance, the clearing data appear convincingly especially through the extensive comparison to other clearing approaches and methodological variations. The immunostaining data are very impressive and provide a lot of novelty. Thus, overall this study can have a lot of impact.

However, there are a few issues which need further exploration and explanation.

1. I am wondering about the limits of this approach. 500um is not a very challenging block size. Other studies offered more, also on human tissue. Given the claims that this method should be feasible to reconstruct whole brain connectomics this block size is still a long way from reconstructing a whole telencephalon, even in a mouse, to be realistic. Also, the authors should speculate about feasible antibodies to visualize axonal connectivity. Post mortem tracing techniques are rare for the human brain and most them are limited to either MR diffusion or lipophilic diffusion tracers, the latter covering only short range connections and not being compatible with this clearing method. So please elaborate on this and give clear reason or perspectives, why you only used 500um slice thickness and how antibodies could work or help with studies on connectivity. To be honest, I achieved already super clearance more than 30 years ago in 400um sections by simply filling cells post mortem with lucifer yellow and emersing them in DMSO but never published these results in terms of clearing, because we still had to deal with sections and didn't even have access to confocal microscopy at that time. So please elaborate on a clear perspective for thicker materials. Talking about resolution, your cellular pictures still look very blurred. Normally parvalbumin stained cells look fantastic in their details. Could you please discuss the limitations of your methods as compared to standard confocal microscopy and show more neurons in a detailed 3d reconstruction and indicate your advantage.

2. In the sections you show e.g. in figure 5a, the staining looks somewhat inhomogeneous, is this due to real biological conditions or just artificial due to your antibody penetration? I am impressed with the results but this needs at least some comments.

3. I have a global question, because I did not see it answered in the Ms as well as in the Suppl.- the answer: do I see individual slices or a full reconstruction?

4. Last, I have a question about the statistics: it is unclear to me on how many different specimen and on how many different subjects these studies were conducted, please make this more transparent. In some instances there is a mention of $n=4$, tested with a t-test: how valid was the test for this? Please make the whole statistics more transparent.

Reviewer #2 (Remarks to the Author):

Dr. Costantini and colleagues present an extension of the original SWITCH method by Murray et al. (2015) specialised on human brain tissue, which improves of SNR of immunofluorescent labels by removing autofluorescent background and adding an antigen retrieval step. They show convincingly, that this method can be used robustly with multiple antibodies on 0.5 mm thick formalin-fixed adult human brain tissue.

Such a robust way for clearing and labelling human brain tissue with antibodies is considered of great general interest to the neuroscience community. Especially the multi-round labelling seems highly useful for a characterisation of the human brain as comprehensive as possible. The successful co-staining of a general neuron marker and GAD67 to distinguish inhibitory neurons

could prove to be highly relevant in the field of cytoarchitectonics to e.g. derive excitation/inhibition ratios over large portions of the human cortex. Finally, the discovery of the autofluorescence-quenching effect of very high concentrations of H₂O₂ are also deemed useful for the wider clearing community, as this step could be easily implemented in many other existing clearing methods.

Nevertheless, I have several comments regarding the work in this manuscript, which are, however, mostly minor in nature:

1. In the method section it is mentioned that 5 donor brains were used, but supplementary table 2 only lists 3 donors.

2. It would have been interesting to see if very dense epitopes such as dendritic or axonal fibre populations are also successfully labelled with the SHORT approach, or to know if such labels have been tested by the authors. However, I understand that this is not the focus of the work.

3. The authors claim in the discussion (line 325) "Also, we implemented a new pipeline compatible for long-term tissue storage of such precious samples." In order to be suitable for long-term storage, the fluorescent signal of the samples must be retained for a long time, which the authors do not show in this manuscript. If other work demonstrates this property of the TDE solution, in which the samples are kept, it might be interesting for the reader to reference those.

4. In line 56 the authors talk about the possibility to physically expand the tissue and write: "[such methods] allow increasing the sample mesh size with a decrawling effect". I have no idea what they mean by this effect. This could be explained to the reader. In line 77 a dot appears to be missing between "LSFM" and "In particular". In line 100-101 it says "This process transforms the tissue into a heat and chemical hybrid". I assume the authors mean heat- and chemically resistant or something similar to that?

5. It is not fully clear to me why the particular thickness of 500 μm was chosen. Was this purely because of technical limitations, such as clearing capacity (white matter in some samples seems very opaque), label penetration, and/or light penetration of the lower wavelengths labels, or was there an anatomical consideration behind this? For both clearing and labelling of human brain tissue, there have been several publications which show processing of thicker samples e.g., Liebmann et al. 2016, Morawski et al. 2017, Hildebrand et al. 2019, and Zhao et al. 2020. If 500 μm is approaching the limit in terms of tissue thickness for this technique, this should be mentioned to give anyone who wants to use the method a better estimation of how this method might scale up.

6. Given the notoriously fragile nature of these hydrogel samples, it is not immediately clear how the sandwich holder "allowed reducing the distortions of the sectioning process and preserving the tissue characteristics for multi-round cycles" as the samples apparently have to be removed from the holder for multi-round labelling and tissue deformations are expected to occur during that handling. As the tissue seems to be simply "clamped" into place between the two cover slips and not glued into position, it is not apparent how the tissue can be repeatedly mounted without tissue deformations occurring.

7. Lastly, the authors claim their method "could be a key technology to map large human brain regions, up to the whole organ". While I agree that this seems to be the general goal the field is aspiring to, I missed a proper evaluation in the discussion of further hurdles to achieve this. The authors image 500 μm thick sections of a few cm in lateral extent. It would be very interesting for the reader to know, how much data such a comparatively small piece of human brain tissue relates to at the 3,3 μm isotropic resolution used in this manuscript. Especially the multi-label + multi-round data-size would be highly interesting to know, as this would provide a much better impression regarding the scalability of the method when the ultimate aim is to image large parts of or even the entire human brain. Related to this, it would be interesting to discuss how size-adjustable the sandwich imaging chambers are and what would be the maximum sample size the chambers and/or microscope set-up allows for.

Reviewer #3 (Remarks to the Author):

In the manuscript entitled "3D molecular phenotyping of cleared human brain tissues with light-sheet fluorescence microscopy", Luca Pesce et al. describes a 3D molecular phenotyping method (named SHORT) based on standard histological treatments and clearing procedure of the human brain. It is claimed that SHORT allows the 3D multiple molecular characterization on human brain. However, this method is merely an extension application of SHIELD and SWITCH. The imaging results shown in this manuscript are not satisfied. More importantly, the current data are not sufficient to support the claim and conclusion. It also lacks convincing results in validity.

The following are some questions and suggestions about this manuscript:

1. The main selling point of SHORT was 3D characterization of the human brain with a combination of multicolors and multi-rounds labeling. However, SWITCH (doi.org/10.1016/j.cell.2015.11.025) has realized similar results in 100 μm human slices with dozen cycles. In this manuscript, only the thickness has been increased to 500 μm , which is only due to the increase in labeling time. In addition, SHORT can only achieve three rounds labeling. I don't think it is innovative enough to be published in this journal.
2. The manuscript claims to have performed a 3D molecular phenotype of human brain. However, the depth is only discussed in Figure 3a and other places only show stacked images. However, according to my experience, the density of stacked images with a thickness of 500 μm for various types of cells is much higher than the results shown in the manuscript. Is this because there is no uniform labeling inside the tissue? More 3D depth information should be provided.
3. As for the multi-rounds labeling, the loss of information between different rounds is an important aspect. In other words, the protein loss during tissue processing needs to be within an acceptable range. However, from the results shown in Figure 3d, the protein loss has been very serious since the second round. The result indicates that in addition to the first round, the validity of results in other rounds needs further proof.
4. There are reports that the sample will undergo significant deformation after being treated with SDS. The tissue will shrink to a certain extent after using TDE, which is also indicated in the manuscript. However, the authors did not conduct a quantitative analysis of tissue deformation. Furthermore, the specimens were delabeled with SDS-containing clearing solution at 80°C. The authors also did not quantify the impact of this process on tissue deformation. This will lead to inaccurate results of multiple rounds of staining

We thank the editors and reviewers for their thorough and careful evaluation of our manuscript. We have found all the comments very useful while revising our paper. In the marked-up version of the manuscript the amends are in tracked changes (in red). Below follows our point-by-point answers to each comment.

Reviewer #1 (Remarks to the Author)

In the present manuscript the authors describe an optimized clearing approach (SHORT) which is based on the earlier clearing protocol SWITCH. They apply this to thick sections of human cortical tissue and combine the clearing multicolor immunostaining and visualize the clearing and labelling results using light sheet fluorescence microscopy.

At first glance, the clearing data appear convincingly especially through the extensive comparison to other clearing approaches and methodological variations. The immunostaining data are very impressive and provide a lot of novelty. Thus, overall this study can have a lot of impact.

However, there are a few issues which need further exploration and explanation.

1.

A. I am wondering about the limits of this approach. 500um is not a very challenging block size. Other studies offered more, also on human tissue. Given the claims that this method should be feasible to reconstruct whole brain connectomics this block size is still a long way from reconstructing a whole telencephalon, even in a mouse, to be realistic.

We agree with the reviewer that for small laboratory animals' organs there are very successful methods that allow to reach the transparency of an entire brain, indeed, in our laboratory we are using various of them as shown in a recent publication (Silvestri et al. "Universal autofocus for quantitative volumetric microscopy of whole mouse brains" Nat. Met 2021). However, human brain clearing presents additional challenges, such as variability of post-mortem fixation conditions, presence of blood inside the vessels, and autofluorescence signals coming from lipofuscin-type pigments, that make it very difficult to clear in comparison to organs derived from other species used in research. In the last few years, various approaches have been proposed to clear human brain specimens, however, they present some limitations. Most of them are compatible with specific samples such as fresh-frozen samples, fetal brains, specimens with controlled post-mortem conditions perfusion fixation. Also, the immunostaining is usually performed with few antibodies and in thinner sections. The advent of tissue-hydrogel engineering methodologies has extended the utility and labeling efficiency in human specimens. In particular, ELAST converts the tissue into an elastic hydrogel allowing homogenous staining of a 1 cm-thick section with several antibodies. However, the sample preparation requires long processing times and advanced custom-made equipment.

Concerning the classic version of SWITCH, it uses 100- μ m human slab for performing multilabeling, without showing the probe penetration in thicker samples. SHIELD shows an efficient probe penetration in a 1-mm human slab for only two markers (calretinin and GFAP). Such works demonstrated that the staining for more than two markers was successfully performed on samples of 100 μ m-thick, and many antibodies can fail to work following tissue transformation protocol procedures and clearing process. For instance, CLARITY procedure in humans strongly reduces the NeuN staining efficiency as demonstrated in the work of Scardigli et al 2021 (doi.org/10.3389/fnana.2021.752234).

Here we compared SHORT with SHIELD clearing and we demonstrated that latter is characterized by a dis-homogenous staining of high-density epitopes such as NeuN in 500 μ m SHIELD-

processed slices of human cortex due to low probe penetration (Fig. 1d, Supplementary Figs. 2, 3).

In contrast, SHORT allows multiple staining of slabs up to 500- μm thickness increasing by 5 times the imaging depth in both grey and white matter.

To clarify these points further, we added text to the introduction in which we described the limitation of the previous clearing methods and the improvement introduced with SHORT.

“Recent applications of optical clearing methods to the human brain provide new details for structure-function relationships in healthy and pathological conditions. However, such methodologies have some drawbacks and limitations. Most of them are compatible with specific samples such as fresh-frozen samples, fetal brains, specimens with controlled post-mortem conditions. Also, the immunostaining is usually performed with small dyes/few antibodies and in thinner sections. The advent of the tissue-hydrogel engineering methodologies has extended the utility and labeling efficiency in human samples. In particular ELAST converts the tissue into an elastic hydrogel allowing homogenous staining of a 1 cm-thick section with several antibodies. However, the sample preparation requires long processing times and advanced custom-made equipment.”

“Also, the time extension for achieving sufficient transparency can affect the staining efficiency, and several antibodies can fail to work following a tissue transformation protocol in the grey as well as white matter. For instance, CLARITY procedures strongly reduce the staining efficiency for specific epitopes in human formalin-fixed specimens (Scardigli et al 2021). Due to these problems, large-scale reconstruction in the white and grey matter of different neuronal markers and cell types with molecular details remains an unmet goal in the human brain.”

Finally, to demonstrate the capability of using SHORT to reconstruct volumetric portions of the human brain, we added in the main text the reference to an open access repository (DANDI), where to find the datasets of a portion of Broca’s area. By cutting the area in 16 slices of 500 μm of thickness it was possible to clear, label and image a volume of 1.5x1.3x0.8 cm^3 with LSM. This work is the subject of a different publication that is now available as preprint on biorxiv: <https://doi.org/10.1101/2021.10.20.464979>.

“With SHORT volumetric blocks of the human brain can be reconstructed by cutting the area of interest in slices of 500 μm of thickness. In this paper, we showed the reconstruction of 500 μm -thick slices of various areas of the brain demonstrating the versatility of the method. SHORT was applied in Costantini et al. 2021 (ref .<https://www.biorxiv.org/content/10.1101/2021.10.20.464979v1>) demonstrating how this approach can be used to reconstruct a portion of Broca’s area of 1.5x1.3x0.8 cm^3 cutting it in only 16 slices allowing the cell census of this volumetric portion of the cortex. The entire dataset is available on the DANDI archive (<https://gui.dandiarchive.org/#/dandiset/000026>).“

B. Also, the authors should speculate about feasible antibodies to visualize axonal connectivity. Post mortem tracing techniques are rare for the human brain and most them are limited to either MR diffusion or lipophilic diffusion tracers, the latter covering only short range connections and not being compatible with this clearing method. So please elaborate on this and give clear reason or perspectives, why you only used 500um slice thickness and how antibodies could work or help with studies on connectivity. To be honest, I achieved already super clearance more than 30 years ago in 400um sections by simply filling cells post mortem with lucifer yellow and emerging them in DMSO but never published these results in terms of clearing, because we still had to deal with sections and didn’t even have access to confocal microscopy at that time. So please elaborate on a clear perspective for thicker materials.

Concerning the possibility of studying connectivity with clearing approaches we agree with the

reviewer that it could be very interesting and would open the possibility of answering several unsolved questions. Although this was not the aim of our study, we tried to combine our clearing method with axon/myelin-specific staining. We tested the compatibility of the lipophilic dye DiD with SHORT. We successfully replicated the result obtained by SWITCH (applied in mouse samples) reaching reliable staining up to 100 μm in formalin-fixed human brain specimens. These preliminary results suggest that myelin staining can be achieved with SHORT. Nevertheless, the compactness of the axon in the white matter makes it difficult to obtain a good fluorescence contrast in depth, however, we are confident that with further investigation and with specific optimization, deeper labeling can be accomplished.

We added these preliminary results in the supplementary materials (Supplementary Fig. 7) and we added a paragraph in the Discussion underlining the importance of further investigation in the combination of clearing techniques with axonal staining for large-scale high-resolution connectomics analysis.

Discussion:

"...We also performed staining using DiD on 100 μm -thick samples of superior frontal cortex, demonstrating the compatibility of SHORT with lipophilic tracing for the rapid visualization of human cytoarchitecture using other methods than immunohistochemistry. The advantages of SHORT may allow combining molecular details by antibody staining and cellular connectivity with lipophilic dyes from diverse cell types within a single tissue."

C. Talking about resolution, your cellular pictures still look very blurred. Normally parvalbumin stained cells look fantastic in their details. Could you please discuss the limitations of your methods as compared to standard confocal microscopy and show more neurons in a detailed 3d reconstruction and indicate your advantage.

To show the high-resolution achievable with our LSFM ($0.55 \times 0.55 \times 3.3 \mu\text{m}^3$) we added additional images in the Results (see Supplementary Fig. 5 and Supplementary Movie 1-3 that show high-resolution images obtained with LSFM of several neuronal and non-neuronal markers). The blurred effect is attributable to the downscaling process used for the representation of the fused reconstruction as discussed in the Results. Finally, we added a paragraph in the Discussion that compares confocal microscopy to LSFM microscopy.

Supplementary Figure 5. High-resolution images acquired by LSFM (resolution of $0.55 \times 0.55 \times 3.3 \mu\text{m}$). Representative images showing morphological details of calretinin (CR)-, glutamic acid decarboxylase 67 (GAD67)-, vasoactive intestinal peptide (VIP)-, somatostatin (SST)-, calbindin

(CB)-, parvalbumin (PV), neuronal nuclear protein (NeuN)-, glial fibrillary acidic protein (GFAP)-immunoreactive cortical neurons, and vimentin (VIM) immunolabeling of blood vessels. Scale bar 100 μm for all images, except for GFAP (scale bar 10 μm).

Discussion:

“Although LSFM lacks axial resolution power, it is the best optical imaging technique for probing multiple markers in large specimens. Indeed, LSFM offers fast acquisition time and low photobleaching with respect to other conventional optical techniques, such as confocal and two-photon fluorescence microscopy”.

2. In the sections you show e.g. in figure 5a, the staining looks somewhat inhomogeneous, is this due to real biological conditions or just artificial due to your antibody penetration? I am impressed with the results but this needs at least some comments.

With respect to Figure 5a, LSFM imaging suffers illumination artifacts, in particular shadowing effects. When an obstacle blocks the light at one point, the tissue behind it is darkened. To perform the acquisition, the sandwich-sample holders are completely immersed in a medium with a high-refractive index (91% glycerol or 68% TDE, total volume 800 ml to fill the microscope and sample plexiglass chamber) which reduces the optical aberration. However, during the acquisitions, some bubbles can remain attached to the surface of the glass and create a dark shadow. We think that this is the case of the second acquisition in figure 5a (CR-SST-NeuN). We, therefore, cut and removed the part of the image with the problem to avoid confusion on why that part of the tissue is darker with respect to the rest of the slab.

Fig. 5 SHORT allows multiple rounds of staining, imaging, and stripping of large portions of human brain slices.

3. I have a global question, because I did not see it answered in the Ms as well as in the Suppl.-the answer: do I see individual slices or a full reconstruction?

To specify what we are showing in all the figures we added in the caption the precise number of slices used to obtain the maximum intensity projection (MIP) of each reconstruction. To avoid confusion, we did not perform the MIP on all the slices of the volumetric reconstructions but only on a few of them, ≈ 20 -150 slices for each figure, depending on how crowded was the labelling. Moreover, to show the 3D data better, we added additional figures in the supplementary materials. In Supplementary Figure 14, we inserted the XZ and XY MIP of 3 different neuronal markers (SST, VIP, and CR) to show the homogenous staining along with the thickness. Also, we added 4 videos of 1) a high-resolution stack of CR (Supplementary Movie 1), SST (Supplementary Movie 2), NeuN (Supplementary Movie 3), and 2) downsampled reconstruction of SHORT-processed slices (Supplementary Movie 4). Finally, we added in the main text (both Material and Result sections) the information on how the figures were prepared.

“Supplementary Figure 14. Multiplexed staining of SHORT-processed slice and the homogenous staining along the thickness. xy and zy MIP images of the superior frontal cortex stained for SST (Alexa Fluor 568; orange), VIP (Alexa Fluor 647; blue), and CR (Alexa Fluor 488;

grey). MIPs of 150 slices (whole thickness). Scale bar xy images: 1 cm; scale bar xz = 500 μm ; scale bar of the magnified insets = 100 μm .

4. Last, I have a question about the statistics: it is unclear to me on how many different specimen and on how many different subjects these studies were conducted, please make this more transparent. In some instances there is a mention of $n=4$, tested with a t-test: how valid was the test for this? Please make the whole statistics more transparent.

To quantify the autofluorescence signal in each channel, consecutive small pieces of cerebral cortex (precentral cortex) from patient "sample 1" (Table S2) were treated with the different autofluorescence reagents (hydrogen peroxide (H_2O_2), Quadrol, ascorbic acid, hydroquinone, sodium borohydride (NaBH_4), copper sulfate (CuSO_4), and Sudan Black (SD) and tissue transformation protocol (SWITCH and SHIELD) and then acquired at 405, 488, 561 and 638 nm (laser power 5 mW). Because the accumulation of LF increases during aging and leads to an increase of the autofluorescence signals, we decided to characterize the autofluorescence reagent's effect on the brain of a very old patient (99 years old) that presented the more severe presence of these pigments. To do that, using Fiji, we randomly selected 4 ROIs of $500 \times 500 \mu\text{m}^2$ in the grey matter of volumetric reconstruction of $\approx 5 \times 5 \times 0.5 \text{ mm}^3$ and we analyzed the fluorescence signals in all 4 channels. Next, we calculated means, standard deviations, and t-test. We added detailed information in the Methods section.

"Data and Statistical Analysis:

For tissue autofluorescence characterization, cleared samples were imaged at 405, 488, 561, and 638 nm using our custom-made LSM. To quantify the fluorescence intensities in each channel, 4 regions of interest (ROI) of $500 \times 500 \mu\text{m}$ were randomly selected in the grey matter of volumetric reconstruction of $\approx 5 \times 5 \times 0.5 \text{ mm}^3$, using Fiji [55]. The 4 ROIs were analyzed for each treatment and tissue transformation protocol on consecutive small pieces of cortex (precentral cortex) from patient "sample 1" (Table S2). We decided to characterize the effect of the various reagents on the brain of a very old patient (99 years old) as it presented the more severe presence of lipofuscin, whose accumulation increases during aging leading to an enhancement of the autofluorescence signals. The laser power at 405, 488, 561 and 638 nm was set at 5 mW. Means and standard deviations were plotted using Origin (OriginLab Corporation, Origin 2019b). The significance of the difference in mean values was determined using the two samples t-test. The statistical significance was set at an α level = 0.05 (: $P < 0.05$; **: $P < 0.01$; ***: $P < 0.001$; ****: $P < 0.0001$). To guarantee the reproducibility of the findings, 4 independent experiments were performed on different slices".*

Reviewer #2 (Remarks to the Author):

Dr. Costantini and colleagues present an extension of the original SWITCH method by Murray et al. (2015) specialised on human brain tissue, which improves of SNR of immunofluorescent labels by removing autofluorescent background and adding an antigen retrieval step. They show convincingly, that this method can be used robustly with multiple antibodies on 0.5 mm thick formalin-fixed adult human brain tissue.

Such a robust way for clearing and labelling human brain tissue with antibodies is considered of great general interest to the neuroscience community. Especially the multi-round labelling seems highly useful for a characterisation of the human brain as comprehensive as possible. The successful co-staining of a general neuron marker and GAD67 to distinguish inhibitory neurons

could prove to be highly relevant in the field of cytoarchitectonics to e.g. derive excitation/inhibition ratios over large portions of the human cortex. Finally, the discovery of the autofluorescence-quenching effect of very high concentrations of H₂O₂ are also deemed useful for the wider clearing community, as this step could be easily implemented in many other existing clearing methods.

Nevertheless, I have several comments regarding the work in this manuscript, which are, however, mostly minor in nature:

1. In the method section it is mentioned that 5 donor brains were used, but supplementary table 2 only lists 3 donors.

There were, indeed, 3 donors not 5; we changed the number of donors in the main text.

“Brain tissue specimens were collected from 3 different human donors with no known neuropsychiatric disorders that were obtained from the body donation program of the Université de Tours and from the Massachusetts General Hospital (MGH)”.

2. It would have been interesting to see if very dense epitopes such as dendritic or axonal fibre populations are also successfully labelled with the SHORT approach, or to know if such labels have been tested by the authors. However, I understand that this is not the focus of the work.

We tried to combine our clearing method with axon/myelin lipophilic dye DiD staining (see also response 1b to Reviewer 1). We successfully obtained reliable staining of the tissue up to 100 µm. These preliminary results suggest that myelin staining can be achieved with SHORT. Nevertheless, the compactness of the axon in the white matter makes difficult to obtain sufficient fluorescence contrast in depth, however, we are confident that with further investigation and specific optimization deeper labeling can be obtained. We added these preliminary results in the supplementary materials, and we added a paragraph in the Discussion underlining the importance of further investigation in the combination of clearing techniques with axonal staining for large-scale high-resolution connectomics analysis.

“Supplementary Figure 7. DiD staining on a SHORT-processed human brain slice. Downscaled images of the precentral cortex (100 μm-thick; MIP image of 30 slices; depth: 100 μm) and the intensity signal of the fiber stained with DiD. SHORT is compatible with lipophilic dye and the signal along the thickness is sufficient to detect the labeled fibers. Excitation light, 638 nm; laser power, 1 mW. Scale bar = 10 μm”.

Methods

“Lipophilic dye staining with DiD. Slices (100 μm-thick) of superior frontal cortex from Sample 2 were processed with SHORT and then incubated in 0.5% DiD (1,1'-dioctadecyl-3,3,3',3'-tetramethylindodicarbocyanine, 4-chlorobenzenesulfonate salt) (stock solution 1 mg/ml, dilution 1:2) dissolved in 10% SDS (stock solution 20%, dilution 1:2) at 37 °C for 2-3 days. Samples were then soaked in a solution of 0.5% DiD in PBST and incubated at 37 °C with gentle shaking for 1 day. Finally, the samples were incubated with 30% TDE/PBS (v/v) and 68% TDE/PBS (v/v) in gentle shaking to remove the excess of DiD and match the refractive index”.

3. The authors claim in the discussion (line 325) “Also, we implemented a new pipeline compatible for long-term tissue storage of such precious samples.” In order to be suitable for long-term storage, the fluorescent signal of the samples must be retained for a long time, which the authors do not show in this manuscript. If other work demonstrates this property of the TDE solution, in which the samples are kept, it might be interesting for the reader to reference those.

TDE fluorescence preservation of endogenous GFP fluorescence was previously demonstrated up to 2 months (Costantini et al. 2015), and we added the reference in the main text. In addition, to give a specific characterization of our method and demonstrate the compatibility with long-term storage, we performed a new acquisition on a Broca’s area sample stored in the sandwich holder for a year. To verify how much of the reduction was due to the storage or due to the

photobleaching of the fluorophores from multiple acquisitions, we performed two subsequent reconstructions of the same sample, one after a year from the first imaging, and the other after one year and a week. We did not notice a difference between the second and the third acquisition, suggesting that the reduction of the fluorescence intensity detected after a year is mostly linked to the sample storage. We observed a decrease of the fluorescence intensity of 2 times for Alexa Fluor 488 ($P < 0.01$), while for Alexa Fluor 568 and Alexa Fluor 647 the reduction was of 1.3 and 1.2 times, respectively, which was not significant. Nevertheless, in all three channels the retained signal makes possible the consecutive acquisition after prolonged storage, indicating that even after a year of storage the markers are well detectable. We discuss the results obtained in the main text and we show the graphs in Supplementary Fig. 12.

Supplementary Figure 12. Signal retention (signal/background) after 1 year in the sandwich holder. Quantification of the signal retention of 10 neurons acquired after immunostaining of CR (green), SST (yellow), and NeuN (red) (Fig. 4l-n), after a 1 year acquisition, and after a 1 year+1 week acquisition. The sample was kept in the sandwich holder for storage. The data show a reduction of the fluorescence signal of 51% for Alexa Fluor 488 ($P < 0.01$), 24% for Alexa Fluor 568 (not significant), and 14% for Alexa Fluor 647 ($P < 0.05$) after 1 year. No difference between the 1 year acquisition and the 1 year+1 week acquisition. MIP images (slice number 20; depth: 70 μm). Scale bar $xy = 1000 \mu\text{m}$; scale bar $x = 500 \mu\text{m}$; scale bar $yz = 250 \mu\text{m}$. The magnified image below the downscaled reconstruction highlights that after a year of storage the markers are well

detectable. Scale bar = 100 μm .

Methods:

“For signal retention, 10 neurons CR-, SST-, and NeuN-immunoreactive in a region of 500 x 500 μm^2 were selected in the following downsampled reconstructions: after the immunostaining process (column 1; mean and standard deviation, SD), after 1 year in the sandwich holder (column 2; mean and SD), and then reacquired after another week for the quantification of the bleaching process (column 3; mean and SD). The signal retention was calculated as signal to background by taking 10 ROIs corresponding to the cellular soma (10 x 10 μm^2) and 10 ROIs of the background (10 x 10 μm^2) using Fiji, and plotted with Origin”.

4. In line 56 the authors talk about the possibility to physically expand the tissue and write: “[such methods] allow increasing the sample mesh size with a decrawling effect”. I have no idea what they mean by this effect. This could be explained to the reader. In line 77 a dot appears to be missing between “LSFM” and “In particular”. In line 100-101 it says “This process transforms the tissue into a heat and chemical hybrid”. I assume the authors mean heat- and chemically resistant or something similar to that?

We corrected these issues and clarified the sentence in line 56.

“Additionally, the recent advent of expansion microscopy (15) and its variants (..) grants super-resolution imaging by expanding the hybrid sample-hydrogel, increasing the sample mesh size and improving the accessibility and density of labeling for proteins even within a dense specimen region (2, 15).

We rephrased the sentence lines 100-101:

“This process transforms the specimen into a heat and chemical resistant hybrid.”

5. It is not fully clear to me why the particular thickness of 500 μm was chosen. Was this purely because of technical limitations, such as clearing capacity (white matter in some samples seems very opaque), label penetration, and/or light penetration of the lower wavelengths labels, or was there an anatomical consideration behind this? For both clearing and labelling of human brain tissue, there have been several publications which show processing of thicker samples e.g., Liebmann et al. 2016, Morawski et al. 2017, Hildebrand et al. 2019, and Zhao et al. 2020. If 500 μm is approaching the limit in terms of tissue thickness for this technique, this should be mentioned to give anyone who wants to use the method a better estimation of how this method might scale up.

In the last few years different methods have been proposed to perform human tissue clearing, however, they have some limitations. In particular, one of the main disadvantages is that they used controlled post mortem fixation conditions that prevent their application to biobank samples that are usually storing tissues in formalin for years. For example, Liebmann et al. 2016 used only fresh-frozen samples, Morawski et al. 2017 6-week formalin-fixed blocks, and Zhao et al. 2020 in situ-controlled perfusion-fixation of whole brains. Another frequent limitation is the compatibility of the clearing procedures with antibody staining since the labeling process efficiency decreases after the delipidation process. The MASH method proposed by Hildebrand et al. (2019) is compatible only with a small range of exogenous dyes: acridine orange, methylene blue, methyl green, neutral red, and nuclear staining and lacks to give information on the staining efficiency using antibodies (as described in the discussion of their work).

In our paper we wanted to find a clearing protocol suitable with most specimens, even for samples

formalin-fixed over years, compatible with antibody staining. To describe the limitation of the published clearing methods better and the improvement introduced with SHORT we added a paragraph in the Introduction (see also response 1a to reviewer 1).

Concerning the decision of using 500- μm of thickness, we evaluated the staining efficiency in the white and grey matter, and we assessed the 500- μm thickness as a limit for obtaining homogenous labeling for different antibody staining (both neuronal, glial, and vasculature). This result increases by 5 times the slice thickness treated with the classic SWITCH technique, allowing to reduce the cutting artifacts and the acquisition times.

6. Given the notoriously fragile nature of these hydrogel samples, it is not immediately clear how the sandwich holder “allowed reducing the distortions of the sectioning process and preserving the tissue characteristics for multi-round cycles” as the samples apparently have to be removed from the holder for multi-round labelling and tissue deformations are expected to occur during that handling. As the tissue seems to be simply “clamped” into place between the two cover slips and not glued into position, it is not apparent how the tissue can be repeatedly mounted without tissue deformations occurring.

The sandwich holder allows reducing mounting distortion that can be present when the tissue is just bathed inside a solution and can float during the acquisition. Moreover, it flattens the surface of the slab facilitating image acquisition. However, during the multi-round staining, mounting, and unmounting the sample can introduce in-plane tissue distortion. We, therefore, rephrase the sentence in:

“...our sample “sandwich” holder allowed us to facilitate sample mounting and obtaining human slices with a flat surface; moreover, it permits long-term storage of such specimens”.

7. Lastly, the authors claim their method “could be a key technology to map large human brain regions, up to the whole organ”. While I agree that this seems to be the general goal the field is aspiring to, I missed a proper evaluation in the discussion of further hurdles to achieve this. The authors image 500 μm thick sections of a few cm in lateral extent. It would be very interesting for the reader to know, how much data such a comparatively small piece of human brain tissue relates to at the 3,3 μm isotropic resolution used in this manuscript. Especially the multi-label + multi-round data-size would be highly interesting to know, as this would provide a much better impression regarding the scalability of the method when the ultimate aim is to image large parts of or even the entire human brain. Related to this, it would be interesting to discuss how size-adjustable the sandwich imaging chambers are and what would be the maximum sample size the chambers and/or microscope set-up allows for.

Big Data generation, storage, and analysis are indeed key aspects of LSFM imaging. In parallel to tissue preparation, optimization of data management needs to progress as well. We are writing a different paper specifically focused on the LSFM apparatus and all the software created and used to run the acquisitions to give all the information related to the topic. Nevertheless, we do agree that information about the data size and time of acquisition are important to provide an idea of the difficulties that this type of microscopy encounters, as high-resolution imaging of large samples requires computational approaches that tackle downstream challenges in large-scale data analysis and management. We added text to the methods in which we described an example of acquisition and data storage.

“As example: the acquisition of a 500- μm thick, slice of a Broca’s area of $2 \times 2 \text{ cm}^2$ stained with 2 different dyes requires 30 minutes and generates 0.4 terabyte. To efficiently store the 3D subvolumes (termed tiles) acquired by LSFM, the files were compressed to JPEG2000 format,

which allows a reduction of the large-scale data by 60 times. To process and extract biological meaning from the acquired image data, the tiled raw images were resliced and downscaled with an isotropic resolution of $3.3 \times 3.3 \times 3.3 \mu\text{m}^3$ (starting resolution: $0.5 \times 0.5 \times 3.3 \mu\text{m}^3$). Next, the multi-tiles downsampled images were fused to perform a downscaled reconstruction of the whole slice. Such image processing allows converting the stained Broca's area of $2 \times 2 \text{ cm}^2$ in a Z-stack of 6 gigabyte (3 for each color) which can be visualized and analyzed by Fiji".

Concerning the scalability of the imaging we confirm that the sandwich holder is size-adjustable. We are buying both the spacer, the holding glass, and the cover glass from companies that produce customizable size items and we are assembling them as presented in the paper. Indeed, in a recent work, we used an $8 \times 8 \text{ cm}^2$ sandwich size (Scardigli et al, 2021; <https://doi.org/10.3389/fnana.2021.752234>) to increase the XY size, demonstrating the versatility of the method. We added the information about the possibility of adjusting the size of the sandwich in the method and discussion sections.

"...Both the spacer, the holding glass, and the cover glass are purchased from companies (Microlaser srl, Sesto Fiorentino, Italy; Laser Optex Inc., Beijing, China) that produce customizable size items permitting to adjust the size of the sandwich according to the application."

"Although our pipeline permits modulating the XY dimension of the samples using appropriate sandwiches that are size-adjustable".

Reviewer #3 (Remarks to the Author):

In the manuscript entitled "3D molecular phenotyping of cleared human brain tissues with light-sheet fluorescence microscopy", Luca Pesce et al. describes a 3D molecular phenotyping method (named SHORT) based on standard histological treatments and clearing procedure of the human brain. It is claimed that SHORT allows the 3D multiple molecular characterization on human brain. However, this method is merely an extension application of SHIELD and SWITCH. The imaging results shown in this manuscript are not satisfied. More importantly, the current data are not sufficient to support the claim and conclusion. It also lacks convincing results in validity.

The following are some questions and suggestions about this manuscript:

1. The main selling point of SHORT was 3D characterization of the human brain with a combination of multicolors and multi-rounds labeling. However, SWITCH (doi.org/10.1016/j.cell.2015.11.025) has realized similar results in $100 \mu\text{m}$ human slices with dozen cycles. In this manuscript, only the thickness has been increased to $500 \mu\text{m}$, which is only due to the increase in labeling time. In addition, SHORT can only achieve three rounds labeling. I don't think it is innovative enough to be published in this journal.

While the SWITCH method obtained a good clearing and labeling in $100 \mu\text{m}$ -thick human brain slices with a dozen cycles, we respectfully disagree with the reviewer that an increase of 5 times of tissue thickness is not innovative enough in human brain tissue clearing.

Working with human brain slices is more challenging than handling commonly used laboratory species due to the variability of post-mortem conditions, intrinsic autofluorescence signal, and tissue transformation treatments that can affect the staining process. We optimized a protocol compatible with several neuronal, vasculature, and glial markers, working with human samples of different ages and formalin fixation times.

A block of 8 mm instead of being cut in 80 slices can be reconstructed with only 16 slabs (as we did in the Costantini et al. 2021; doi: <https://doi.org/10.1101/2021.10.20.464979>) reducing the

cutting artifacts, the tissue preparation time, the imaging time, and the slice alignment problems.

Moreover, concerning the multi-rounds labeling, we reduced the antibody removal time to 4 hours, with respect to 1-2 days of the classic SWITCH version (doi.org/10.1016/j.cell.2015.11.025).

To better explain the innovation of the SHORT method we added a paragraph in the introduction.

2. The manuscript claims to have performed a 3D molecular phenotype of human brain. However, the depth is only discussed in Figure 3a and other places only show stacked images. However, according to my experience, the density of stacked images with a thickness of 500 μm for various types of cells is much higher than the results shown in the manuscript. Is this because there is no uniform labeling inside the tissue? More 3D depth information should be provided.

We realized that we did not show sufficient images of the reconstruction along the Z axis in the work. To address the issue, we added more images and videos to the supplement. In Supplementary Figure 14, we inserted the XZ and XY MIP of 3 different neuronal markers (SST, VIP, and CR) to show the homogenous staining along with the thickness. Also, we added 4 videos of high-resolution stack of the CR, SST, and NeuN immunostainings, and downsampled reconstruction of SHORT-processed slices (Supplementary Movies 1-4).

To specify what we are showing in all the figures we added in the caption the precise number of slices used to obtain the maximum intensity projection (MIP) of each reconstruction. To avoid confusion, we did not perform the MIP on all the slices of the volumetric reconstructions, but only on a few of them, ≈ 20 -150 slices for each figure, depending on how crowded was the labelling.

Finally, we added in the main text (both in the Materials and Results sections) the information on how the figures were prepared. The images shown in Fig. 1c-f, Fig. 3a, Supplementary Figs. 2, 3 represent the whole thickness of the processed tissue as they are the z profile of the slice acquired by the LSM. The apparatus is an inverted microscope that acquired the images with an inclination angle of 45° , therefore the single image represents a 45° inclined z plane of the whole slice: the shown FOV is $\approx 700 \mu\text{m}$ for a $500 \mu\text{m}$ -thick slice.

We used this profile to optimize the staining for the high-density epitope marker NeuN. We quantified the probe penetration of NeuN in SWITCH- and SHIELD-processed slices combined with several immunofluorescence quenching reagents and temperature incubation and we demonstrated that the best combination between the tissue transformation protocol and the quenching reagents was oxygen peroxide combined to antigen retrieval, with the antibody incubation at 37°C . Finally, with the same approach, we demonstrated the not-uniform labeling in SHIELD-processed slices.

3. As for the multi-rounds labeling, the loss of information between different rounds is an important aspect. In other words, the protein loss during tissue processing needs to be within an acceptable range. However, from the results shown in Figure 3d, the protein loss has been very serious since the second round. The result indicates that in addition to the first round, the validity of results in other rounds needs further proof.

This is an important point. The type of analysis that we are proposing with our method is only qualitative. We cannot use SHORT to obtain information on how much a protein is expressed inside the cell of interest, but only if it is present or not. Tissue transformation techniques relying on lipids removal cause protein loss and we believe that this applies not only to SHORT, but also to the other clearing methods.

Nevertheless, to increase the reliability of the obtained results, we raised the number of neurons

(total number 30) considered for the analysis (Figure 3d). The histogram shows a 30% decrease in fluorescence intensity after the first round and another 10% loss after the second round. The signal to noise ratio observed is acceptable and compatible with the immunostaining of different proteins in SHORT-processed slices. Moreover, to demonstrate the signal to noise, we randomly selected 3 ROIs of $750 \times 750 \mu\text{m}^2$ and we counted the neurons in all rounds, finding a correspondence of 100% in the number of neurons in all rounds. This observation demonstrates that using SHORT, it is possible to perform qualitative analysis and multiplexing profiling of several targets in formalin-

fixed tissue.

Results:

“Our data show that the SNR of SST immunostaining in rounds 1, 2, and 3 decreases from round 1 to round 2. This suggests that the amount of protein lost in the stripping process decreases after every round, but antigens are still preserved and can be detected using SHORT. We achieved multiplexed profiling of 7 different targets in the same piece of human superior frontal cortex by multispectral imaging of 3 antigens in each round.”

4. There are reports that the sample will undergo significant deformation after being treated with SDS. The tissue will shrink to a certain extent after using TDE, which is also indicated in the manuscript. However, the authors did not conduct a quantitative analysis of tissue deformation. Furthermore, the specimens were delabeled with SDS-containing clearing solution at 80°C . The authors also did not quantify the impact of this process on tissue deformation. This will lead to inaccurate results of multiple rounds of staining.

Tissue deformations due to SDS lipids removal and TDE clearing were already described in previous reports. SWITCH's original protocol used a SDS clearing temperature of 70°C and SHIELD a temperature up to 55°C as used in this work (there are plenty of other works that use SDS to remove lipids, but these two are the ones more related to our work).

Concerning TDE, Costantini et al. (2015) (doi: 10.1038/srep09808) demonstrated in TEM images the preservation of the ultrastructure after TDE treatment. Subsequent publications successfully used TDE as RI matching medium. The deformations observed using SDS and TDE are linear, therefore they did not influence the ratio between the different compartments of the sample.

In addition, for the stripping process, we reduced the incubation time from 1-2 days to 4 hours at 80°C (a reduction of 12 times). Other works (doi.org/10.1016/j.cell.2015.11.025 and https://doi.org/10.1038/s41467-020-18422-8) used long incubation time (usually overnight) for obtaining an efficient antibody removal, suggesting that SHORT is more conservative.

Nevertheless, we performed an additional analysis to quantify the distortion associated with the

high-temperature incubation, by performing a single cell analysis of SST-immunoreactive neurons before and after a stripping round. We measured SSIM (structure similarity index measure) and the branch tortuosity between stained images for SST after clearing (red) and the stripping process (green) demonstrating a high SSIM index value (0.9 ± 0.06) and no significant structural change and deformation of the sample (Supplementary Fig. 10).

Supplementary Figure 10. Distortion analysis. SHORT-processed human cortical slice immunostained for SST after the clearing process (post-clearing, red images). Then, the antibodies were stripped and restained for SST (stripping, green images). **a** Comparison of the structural similarity between two high-resolution images of a representative SST-immunoreactive neuron after clearing and stripping process using the structural similarity index measure (SSIM index; range value between -1 and 1; two identical images have an SSIM index close to 1). The data are shown as the mean \pm SD ($N = 3$). **b** Comparison of tortuosity of a representative SST branch obtained between the high-resolution images from post-clearing (red) and after stripping (green). The minimum distance (D) and the real branch length (d) between the two ends of each individual branch were measured to evaluate the morphological changes using the torsional index (d/D). The branch alteration after the stripping process is not statically significant (N.S.; $N = 6$). LSFM images; scale bar = $10 \mu\text{m}$; scale bar of the merged image of SST branch = $5 \mu\text{m}$.

Methods

The degree of the deformation of the tissue caused by SHORT during the stripping process was performed on the high-resolution LSFM images, by using the structural similarity index (SSIM index) and the morphological deformation of the branching patterns at single-cell level in SST-immunoreactive neurons. For the SSIM index, 3 different pairs of neurons were cropped and scaled up by a factor of 4 to obtain a minimum number of 256×256 pixels. Then, the images were processed using the plugin SSIM index, to quantify the structural similarity. For the branching pattern tortuosity, the minimum distance and the real branch length of 6 different dendrites (see Fig. S10) was calculated using Fiji and plotted with Origin.

REVIEWERS' COMMENTS:

Reviewer #1 (Remarks to the Author):

Most of my concerns have been met by the author's rebuttal and the respective changes and additions to the manuscript/supplement. There is only one remaining question, namely the applicability of the t-test for the statistical analysis. So, did they test for a normal distribution? Otherwise, this manuscript now adds important new methodological approaches interesting for a broader readership.

Reviewer #2 (Remarks to the Author):

The authors have addressed all my comments extensively. I have no further comments regarding the manuscript at this point and no objections against its publication.

Reviewer #3 (Remarks to the Author):

In the revised manuscript entitled "3D molecular phenotyping of cleared human brain tissues with light-sheet fluorescence microscopy", Luca Pesce et al. made a great improvement. The comments are given as follows:

1. For the multi-rounds labeling, only three rounds are provided, which are insufficient and much less than that of SWITCH.
2. The authors have added a distortion analysis of the cell structure in the revised manuscript. However, this analysis is too limited and should include different scales.

Reviewer #1 (Remarks to the Author):

Most of my concerns have been met by the author's rebuttal and the respective changes and additions to the manuscript/supplement. There is only one remaining question, namely the applicability of the t-test for the statistical analysis. So, did they test for a normal distribution?

Otherwise, this manuscript now adds important new methodological approaches interesting for a broader readership.

We thank the reviewer for this important comment. Our data in Fig. 1 were not normally distributed. We changed the T-test with the Mann-Whitney test (non-parametric test). To make the data analysis more consistent, we improve the number of ROIs for each excitation light (20 ROIs of 20x500 μm^2). The final results do not change the statistic differences, and enhance the properties of our autofluorescence quenching agents tested (Fig. 1).

In addition, we also changed the statistic test (T-test) in the signal retention figure (Supplementary Fig. 12). The statistic differences did not change.

Reviewer #2 (Remarks to the Author):

The authors have addressed all my comments extensively. I have no further comments regarding the manuscript at this point and no objections against its publication.

We thank the reviewer for his evaluation.

Reviewer #3 (Remarks to the Author):

In the revised manuscript entitled “3D molecular phenotyping of cleared human brain tissues with light-sheet fluorescence microscopy”, Luca Pesce et al. made a great improvement. The comments are given as follows:
1. For the multi-rounds labeling, only three rounds are provided, which are insufficient and much less than that of SWITCH. The multi-round strategy requires high tissue preservation, efficient probe penetration, and co-registration of the acquired slices. We tested several antibodies against different neuronal markers, and we found that some of them allow uniform labeling in 500- μm thick during the tissue transformation and multi-round process (7 markers). In addition, our strategy of performing just three rounds allows reducing the post-processing analysis of co-registration. Indeed, the simultaneous acquisition of three different markers for each round makes easier the co-registration of the channels than performing multiple rounds of just one marker. We added the following sentence in the Discussion:

“The simultaneous acquisition of three different markers for each round makes easier the co-registration of different channels in a large portion of the human cortex”.

2. The authors have added a distortion analysis of the cell structure in the revised manuscript. However, this analysis is too limited and should include different scales.

Costantini et al. (2015) (doi: 10.1038/srep09808) demonstrated in TEM images the preservation of the ultrastructure after TDE treatment, and the deformations observed using clearing agent like SDS should not influence the ratio between the different subcellular compartments of the specimens. Also, our system allows fast acquisition with an optical resolution of $\sim 1\mu\text{m}$, which makes out of scope the evaluation of subcellular structure alteration in this work. However, in the future, could be interesting to investigate the distortion of cellular markers (i.e., mitochondria or endoplasmic reticulum) or cytoplasmic proteins using the combination of expansion microscopy techniques such as MAP in combination with high resolution optical techniques.